# (U)NFV: (Un)Supervised Neural Finite Volume Methods for Solving Hyperbolic PDEs

**Nathan Lichtlé**[1,*]**, Alexi Canesse**[1,2,*]**, Zhe Fu**[1,*]**, Hossein Nick Zinat Matin**[1,*]**,**
**Maria Laura Delle Monache**[1]**, Alexandre M. Bayen**[1]

[1]University of California, Berkeley, [2]École polytechnique, LIX, IP Paris, CNRS

{lichtle, alexi.canesse, zhefu, h-matin, mldellemonache, bayen}@berkeley.edu

## Abstract

**We introduce (U)NFV, a modular neural network architecture that generalizes classical finite volume (FV) methods for solving hyperbolic conservation laws.** Hyperbolic partial differential equations (PDEs) are challenging to solve, particularly conservation laws whose physically relevant solutions contain shocks and discontinuities. FV methods are widely used for their mathematical properties: convergence to entropy solutions, flow conservation, or total variation diminishing, but often lack accuracy and flexibility in complex settings. *Neural Finite Volume* addresses these limitations by learning update rules over extended spatial and temporal stencils while preserving conservation structure. It supports both supervised training on solution data (NFV) and unsupervised training via weak-form residual loss (UNFV). Applied to first-order conservation laws, (U)NFV achieves up to **10x lower error** than Godunov's method, outperforms ENO/WENO, and rivals discontinuous Galerkin solvers with lower implementation burden. On traffic modeling problems, both from PDEs and from experimental highway data, (U)NFV captures nonlinear wave dynamics with significantly higher fidelity and scalability than traditional FV approaches. Code, dataset, trained models, and videos can be found at https://nathanlichtle.com/research/nfv.

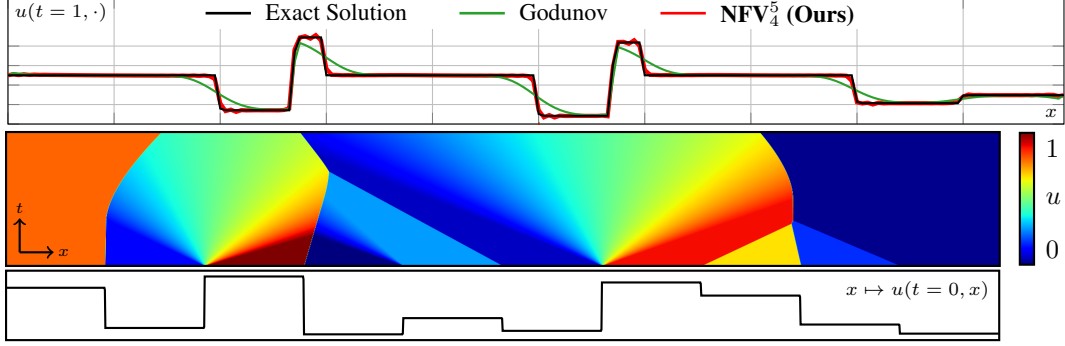

Figure 1: **Prediction of entropy solutions of hyperbolic PDEs. Top:** $\text{NFV}_4^5$ prediction vs. the Godunov scheme for Burgers' equation at a fixed time. LWR predictions follow a similar trend. **Mid:** Entropic solution $u(t, x)$ for Burgers' equation over domain $(t, x) \in [0, 1]^2$. **Bottom:** Corresponding initial condition $u(0, \cdot)$.

# 1 Introduction

*Hyperbolic* partial differential equations (PDEs) are fundamental tools to model propagation and transport phenomena with nonlinear or discontinuous behavior, appearing in areas like fluid dynamics and traffic flow. In this work, we focus on an essential subclass: *conservation laws*, which encode the

---

*Equal contribution

principle that certain physical quantities, such as mass, momentum, or energy, must be preserved over time. A general one-dimensional scalar conservation law takes the form:

$$\partial_t u(x,t) + \partial_x f(u(x,t)) = 0, \tag{1}$$

where $u$ is the conserved quantity and $f$ is the *flux function*.

The solutions of hyperbolic PDEs are difficult to approximate due to discontinuities such as shocks, even when starting from smooth initial conditions (Evans, 2022). Consequently, classical (smooth) solutions typically cease to exist after finite time, and one must instead rely on weak solutions. Closed-form solutions exist only in rare cases, such as on simple Riemann problems LeVeque (2002) or through the Lax-Hopf formula (Lax, 1957; Claudel and Bayen, 2010a;b) in specific concave or convex settings. As a result, most practical applications rely on numerical methods for approximating the PDE solution, with finite volume (FV) methods (LeVeque, 2002) being a popular choice due to their ability to track conserved quantities across discontinuities and capture shock dynamics.

Classical FV methods involve important trade-offs between accuracy near discontinuities, computational cost, stencil size, and implementation complexity. In recent years, neural networks have been explored as flexible and powerful alternatives solvers, showing promise in learning complex dynamics from data or residuals. Yet, many such methods are designed for non-specific models, often at the expense of losing physical structure, including conservation laws and entropy behaviors.

We introduce the *Neural Finite Volume* (NFV) method, a modular architecture tailored to conservation laws, that blends the structure-preserving benefits of FV schemes with the expressiveness of neural networks. Conservation is built into the NFV model, using extended spatial and temporal stencils. We develop both a supervised version, trained on solution data from simple cases, and an unsupervised variant (UNFV), which learns directly from the PDE via a weak-form residual loss. This flexibility allows (U)NFV to adapt to the availability of data, leveraging accurate synthetic or field data when present, or solving directly from the equation when solutions are inaccurate or expensive to obtain. We focus on one-dimensional scalar conservation laws, which are widely used in applications such as traffic flow, pipeline and channel models, and form a standard, well-understood testbed.

**Contributions.** Our main contributions are as follows:

- We propose (U)NFV, a neural architecture that generalizes the structure of finite volume methods and thus preserves conservation properties by construction.
- We introduce two variants: a supervised learning one (NFV) and an unsupervised learning one (UNFV), depending on data availability, using either solution data or a weak-form residual loss.
- We demonstrate strong numerical results on several conservation laws, achieving up to 10x lower error than classical FV solvers, as shown in Figure 1. Additionally, (U)NFV matches the accuracy of discontinuous Galerkin methods, without their mathematical complexity.
- We show that NFV can be trained on field data that does not strictly satisfy the conservation law, and still predicts accurate solutions with more generalizability than classical solvers.

The remainder of the article is organized as follows: Section 2 provides a detailed overview of the related work, Section 3 recalls the FV and introduces necessary notation, Section 4 describes the proposed (U)NFV method in detail, Section 5 presents the experiments and results on hyperbolic PDEs, Section 6 extends the NFV to experimental field highway data, and Section 7 concludes the article. Then, Appendix A provides details about FV schemes, Appendix B illustrates six PDE variants considered in this work, Appendix C expands on the experimental data handling and results from Section 6, and Appendix D details the model architecture, dataset, and hyperparameter choices.

## 2 RELATED WORK

**Numerical methods.** Classical numerical methods for hyperbolic PDEs, such as FV and *discontinuous Galerkin* (DG) (Hu and Shu, 1999) methods, are widely used due to their capabilities in capturing shocks and discontinuities. First-order schemes such as the *Lax-Friedrichs* (Lax, 1954) and *Godunov* (Godunov, 1959b) methods provide robustness but suffer from excessive numerical diffusion, leading to smeared solutions. To address this, higher-order methods like *Essentially Non-Oscillatory* (ENO) (Shu, 1999), *Weighted ENO* (WENO) (Liu et al., 1994), and DG have been introduced, offering improved accuracy in smooth regions while preserving stability near shocks. DG further improves accuracy through local polynomial approximations but incurs high computational

costs (Cockburn and Shu, 1998). In practice, DG and higher-order FV schemes like WENO demand intricate flux constructions, quadrature rules, and stabilization choices, whereas (U)NFV retains FV-like implementation complexity. Despite their accuracy, these methods often require extensive manual tuning effort, motivating the development of flexible, data-driven alternatives.

**NN approaches for PDEs.** Deep learning is a powerful approach for approximating PDE solutions. In the supervised learning case, neural operators such as *Fourier Neural Operator* (FNO) (Li et al., 2020) and *Deep Operator Networks* (DeepONet) (Lu et al., 2021) efficiently approximate solution mappings from parametric inputs, without requiring explicit mesh discretization in the case of FNO. However, these operators have mainly been validated on elliptic or parabolic PDEs, typically characterized by smooth solutions. Conventional neural architectures, such as CNNs (LeCun et al., 1995) for structured domains and GNNs (Bronstein et al., 2017) for irregular geometries, have also been adopted, but supervised models rely heavily on large, high-quality labeled datasets, and often lack intrinsic enforcement of physical constraints, leading to limited generalization and poor accuracy on PDEs involving sharp gradients or shocks (Krishnapriyan et al., 2021).

To reduce data reliance, unsupervised approaches like *Physics-Informed Neural Networks* (PINNs) incorporate PDE residuals directly into training losses (Raissi et al., 2017), proving effective for elliptic and parabolic equations (Raissi et al., 2019; Jagtap et al., 2020). However, PINNs encounter significant difficulties with hyperbolic PDEs, especially in capturing discontinuities and shock dynamics, resulting in unstable optimization, convergence failures, and inaccurate solutions (Wang and Liu, 2021; Fuks and Tchelepi, 2020). Recent variants, such as *Weak PINNs* (wPINNs) (De Ryck et al., 2024), *Parareal PINNs* (PPINNs) (Meng et al., 2020), and *Extended PINNs* (XPINNs) (Jagtap and Karniadakis, 2020), aim to overcome these issues through weak formulations or specialized training strategies. Nonetheless, these adaptations often introduce significant complexity and require extensive hyperparameter tuning, underscoring a need for methods tailored to hyperbolic PDEs.

**NNs for hyperbolic PDEs and conservation laws.** Neural approaches tailored to hyperbolic PDEs have introduced innovations to handle shocks. Weak PINNs (wPINNs) (De Ryck et al., 2024) integrate weak-form residuals or integral constraints to mitigate issues with discontinuities. Others employ neural networks directly within classical FV schemes to learn improved flux reconstructions (Kossaczká et al., 2021; Tong et al., 2024). However, these enhancements typically reintroduce complexity, such as extensive manual parameterization or problem-specific adaptivity, diluting the key advantage of neural flexibility and generality.

Motivated by these limitations, our proposed NFV approach learns local update rules directly from data or PDE residuals. By preserving the fundamental conservation-law structure of traditional FV methods while flexibly leveraging neural networks, NFV achieves significantly higher accuracy, robustness, and scalability with minimal manual intervention.

## 3 Prerequisites and Notations: Finite Volume Methods

Standard FV methods, such as those presented in LeVeque (2002), solve the integral form of the conservation law (1) on a mesh of uniform cells $I_i = [x_{i-1/2}, x_{i+1/2}]$, $i = 1, \cdots, I_{\max}$, with cell length $\Delta x$. The average of $u$ over cell $I_i$ at time $t_n = n\Delta t$, for $n = 1, \cdots, N$ and time discretization $\Delta t$, and the numerical flux through the interface $x_{i+1/2}$ over the time step, are given respectively by

$$u_i^n = \frac{1}{\Delta x} \int_{I_i} u(t_n, x)\, \mathrm{d}x \quad \text{and} \quad F_{i+1/2}^n = \int_{t_n}^{t_{n+1}} f(u(t, x_{i+1/2}))\, \mathrm{d}t. \tag{2}$$

A first-order method $\mathcal{F}$ approximates the numerical flux as $\hat{F}_{i+1/2}^n = \mathcal{F}(u_i^n, u_{i+1}^n)$, while higher-order methods leverage additional cell averages. Let us generalize this framework by including cell averages from previous time steps in order to construct even better approximations. Let $\mathrm{FV}_a^b$ be the class of methods that use a rectangular stencil of $a$ neighboring spatial cells times $b$ past time steps to estimate numerical fluxes. Specifically, define the $(a, b)$-stencil

Figure 2: Example stencil for $\mathrm{FV}_4^2$, taking in a stencil of 2 time steps times 4 space cells.

at interface $i + 1/2$ as $\mathcal{U}_{i+1/2}^n(a, b) = \{u_k^m\}_{k=i+1-\frac{a}{2}, \ldots, i+\frac{a}{2}}^{m=n-b, \ldots, n-1}$, and let an $\mathrm{FV}_a^b$ method $\mathcal{F}$ estimate

numerical fluxes as $\hat{F}_{i+1/2}^n = \mathcal{F}\left(\mathcal{U}_{i+1/2}^n(a, b)\right)$. Classical first-order methods, such as Godunov, fall under class $\text{FV}_2^1$; more details about their computation are provided in Appendix A. To our knowledge, the vast majority of FV methods in the literature use a single time step (i.e., $b = 1$) and a small number of spatial cells. Indeed, designing analytical schemes with larger temporal or spatial stencils becomes exponentially more complex. Finally, the update rule is given by the exact relation

$$u_i^{n+1} = u_i^n - \frac{\Delta t}{\Delta x}\left(F_{i+1/2}^n - F_{i-1/2}^n\right), \tag{3}$$

which in practice is approximated using the numerical fluxes $\hat{F}_{i\pm1/2}^n$, leading to an approximation $\hat{u}_i^n$ of $u_i^n$. Note that the influx of one cell is the outflux of another, which ensures conservation.

## 4    OUR METHOD: NEURAL FINITE VOLUME (NFV)

Our method builds upon the FV framework by using neural networks to approximate the numerical flux. Specifically, we define $\text{NFV}_a^b$ as a generalization of $\text{FV}_a^b$, where the numerical flux $\hat{F}_{i\pm1/2}^n$ is predicted by a neural network $\mathcal{N}$ based on a local $a \times b$ spatiotemporal stencil:

$$\hat{F}_{i\pm1/2}^n = \mathcal{N}(\boldsymbol{u}_{i\pm1/2}^n(a, b))$$

The prediction of the solution is then updated using the classical FV update rule (3), ensuring mass conservation. We explore NFV models ranging from $\text{NFV}_2^1$ (matching Godunov's stencil) to $\text{NFV}_{10}^{11}$, using 11 spatial cells and 11 past time steps – configurations that would be exceedingly complex to design manually due to the high-dimensional stencil involved. This extension enables accurate learning even from noisy field data. In practice, we implement NFV as a CNN (LeCun et al., 1995), which allows efficient computation across stencils due to the vectorized nature of CNNs. Since (U)NFV retains the standard finite volume update, boundary conditions such as Dirichlet, Neumann, or open boundaries can be imposed via ghost cells or prescribed interface fluxes exactly as in classical FV schemes, without modifying the neural architecture.

In all experiments we instantiate $\text{NFV}_a^b$ as a lightweight two-dimensional CNN applied locally on each cell interface: the first layer uses a kernel of size $a$ over the spatial dimension with $b$ input channels (one per time slice), followed by five $1 \times 1$ convolutional layers with 15 channels and either ELU or ReLU activations depending on the flux family. This architecture yields $1105 + 16(ab + 1)$ trainable parameters for $\text{NFV}_a^b$, so even our largest models contain only a few thousand parameters while retaining the exact FV update rule.

We propose two variants of NFV that share the same architecture but differ in their training objectives: the supervised $\text{NFV}_a^b$, trained on reference solutions, and the unsupervised $\text{UNFV}_a^b$, trained directly from the PDE via a weak-form residual loss. The supervised setting applies when solution data is available, while the unsupervised variant enables training when such data is absent, relying instead on the governing conservation laws. Moreover, supervised NFV can also be applied in cases where the PDE is unknown but observational data is accessible, allowing solvers to be deployed directly on field data with only basic physical constraints, such as mass conservation, imposed, and without extensive hyperparameter tuning (see Section 6).

In all our experiments, we therefore train one (U)NFV model per conservation law, and once trained the same network can be applied to many different initial conditions for that equation, so the one-time training cost is largely amortized and in practice remains very short. At inference time, no optimization is solved: each time step is advanced by a single application of the finite volume update rule (3) with numerical fluxes $F_{i\pm1/2}^n$ given by a forward pass of the neural network, so the overall cost of solving an equation scales linearly with the number of time steps.

### 4.1    SUPERVISED LEARNING

Supervised learning offers a straightforward framework for training models when reference solutions are available. In this study, we employ supervised learning not only to approximate the solution of known equations but also to predict field data with unknown governing equations. Although solutions to hyperbolic PDEs are typically defined in the $L_1$ space, we consider their restrictions to

compact subsets where the functions are bounded, thereby allowing treatment within the $L_2$ space. Accordingly, the loss function is defined as the standard mean square error:

$$\mathcal{L}_s = \mathop{\mathbb{E}}_{u_0 \sim \mathcal{R}} ||u - \hat{u}||_2^2$$

where $u$ is the true solution, $\hat{u}$ is the predicted solution, and $\mathcal{R}$ is a distribution over initial conditions.

### 4.2 UNSUPERVISED LEARNING

Unsupervised learning for hyperbolic PDEs is particularly challenging because their solutions often lack closed-form expressions and classical (strong) solutions may not exist. Instead, these equations are typically defined through weak formulations. Although weak solutions are not unique: multiple functions can satisfy the PDE, but only one corresponds to the physically relevant *entropy solution*, which enforces admissibility conditions across shocks and discontinuities.

The unsupervised loss function is defined to minimize the residuals of the weak formulation, in order to approximate the entropy solution. While imposing this loss does not guarantee convergence to the entropy solution, empirical results indicate that our method successfully converges to the entropy solution across various equations and numerous trials. To enhance learning efficiency, we optimize the weak formulation independently at each time step by minimizing the squared residuals. The collection of test functions $\Phi$ consists of 250 randomly sampled, compactly supported polynomials of degree 50 over the spatial domain. The unsupervised loss reads:

$$\mathcal{L}_w = \mathop{\mathbb{E}}_{\substack{\varphi \in \Phi \\ u_0 \sim \mathcal{R}}} \left[ \sum_{n=1}^{N} \left( \sum_{i=1}^{I_{\max}} \left( (\Delta t)^{-1}(\hat{u}_i^n - \hat{u}_i^{n-1}) \int_{I_i} \varphi + f(\hat{u}_i^n)[\varphi]_{x_{i-1/2}}^{x_{i+1/2}} \right) \right)^2 \right]$$

where $\hat{u}_i^n$ denotes the predicted solution at spatial index $i$ and time step $n$, and $\mathcal{R}$ is a distribution over initial conditions. Note that for the scalar conservation laws considered here, integration by parts removes spatial derivatives from the weak-form loss, and time derivatives are handled via finite differences in the FV update, so no explicit spatial derivatives of the primal variables are required during training.

## 5 EXPERIMENTS

Experiments have been designed to answer four main questions:
- Is (U)NFV a compelling alternative to classical finite volume methods?
- Does UNFV converge to an entropy solution despite being trained on the weak formulation?
- How does (U)NFV compare to much more complicated finite element methods?
- Can NFV perform well on field data that contains noise and may not be conservative?

### 5.1 BASELINES

Selecting appropriate baselines for PDE solvers poses challenges due to the diversity in computational frameworks: methods vary by mesh dependency (mesh-free versus mesh-based), solution generation (autoregressive versus single-pass), and generalizability (operator-based versus retrained per initial condition). Therefore, we adopt classical numerical schemes, the foundation of our NFV method, as baselines, ensuring a fair comparison. Given the fact that NFV is developed based on traditional first-order FV methods, the present work provides a compelling case for replacing standard FV solvers with the simpler yet effective NFV method whenever FV methods are typically employed. We consider all the numerical schemes introduced in Section 2 as baselines: first-order FV methods (Godunov, Lax-Friedrichs, and Engquist-Osher), higher-order ones (ENO, WENO), and DG, a finite-element method that is well-known for superior accuracy but suffers from computational burden. More details can be found in Appendix A.

### 5.2 EQUATIONS

**The Lighthill-Whitham-Richards** model (Lighthill and Whitham, 1955; Richards, 1956), known as LWR, is a first-order hyperbolic conservation law used to model traffic flow. It is expressed as

$$\partial_t \rho + \partial_x(\rho v(\rho)) = 0 \tag{4}$$

Table 1: Performance comparison between neural network models and classical numerical schemes. Results are computed over the evaluation set of 1000 piecewise constant initial conditions. For each method, we report mean and standard deviation in $L_2$ norm ($\mathrm{mean}((u - \hat{u})^2)$).

| | 1$^{st}$ order FV | | | | | Higher order FV | | FEM |
|---|---|---|---|---|---|---|---|---|
| | **NFV$_2^1$** | **UNFV$_2^1$** | GD | LF | EO | ENO | WENO | DG |
| G.shields | $\mathbf{1.3e^{-4}_{\pm4e-5}}$ | $2.0e^{-4}_{\pm6e-5}$ | $4.5e^{-4}_{\pm2e-4}$ | $1.3e^{-2}_{\pm4e-3}$ | $4.5e^{-4}_{\pm2e-4}$ | $6.4e^{-4}_{\pm4e-4}$ | $6.4e^{-4}_{\pm4e-4}$ | $3.1e^{-5}_{\pm1e-5}$ |
| Tri. 1 | $\mathbf{1.4e^{-3}_{\pm6e-4}}$ | $1.9e^{-3}_{\pm9e-4}$ | $2.3e^{-3}_{\pm1e-3}$ | $9.6e^{-3}_{\pm4e-3}$ | $2.3e^{-3}_{\pm1e-3}$ | $2.0e^{-3}_{\pm2e-3}$ | $1.9e^{-3}_{\pm2e-3}$ | $2.6e^{-4}_{\pm1e-4}$ |
| Tri. 2 | $\mathbf{2.4e^{-3}_{\pm1e-3}}$ | $3.1e^{-3}_{\pm2e-3}$ | $3.8e^{-3}_{\pm2e-3}$ | $1.4e^{-2}_{\pm8e-3}$ | $3.8e^{-3}_{\pm2e-3}$ | $5.8e^{-3}_{\pm4e-3}$ | $5.8e^{-3}_{\pm4e-3}$ | $4.1e^{-4}_{\pm2e-4}$ |
| Trapez. | $\mathbf{1.1e^{-3}_{\pm4e-4}}$ | $1.6e^{-3}_{\pm7e-4}$ | $2.1e^{-3}_{\pm8e-4}$ | $2.5e^{-2}_{\pm1e-2}$ | $2.1e^{-3}_{\pm8e-4}$ | $6.2e^{-4}_{\pm2e-4}$ | $5.3e^{-4}_{\pm2e-4}$ | $2.9e^{-4}_{\pm1e-4}$ |
| G.berg | $\mathbf{1.4e^{-4}_{\pm9e-5}}$ | $3.8e^{-4}_{\pm2e-4}$ | $4.9e^{-4}_{\pm2e-4}$ | $5.3e^{-3}_{\pm2e-3}$ | $4.9e^{-4}_{\pm2e-4}$ | $1.1e^{-3}_{\pm6e-4}$ | $1.2e^{-3}_{\pm9e-4}$ | $3.4e^{-4}_{\pm2e-3}$ |
| U.wood | $\mathbf{3.8e^{-4}_{\pm1e-4}}$ | $6.9e^{-4}_{\pm2e-4}$ | $9.2e^{-4}_{\pm3e-4}$ | $2.7e^{-2}_{\pm1e-2}$ | $9.2e^{-4}_{\pm3e-4}$ | $1.1e^{-4}_{\pm3e-5}$ | $9.8e^{-5}_{\pm2e-5}$ | $5.9e^{-5}_{\pm2e-5}$ |
| Burgers | $\mathbf{8.5e^{-4}_{\pm3e-4}}$ | $1.3e^{-3}_{\pm6e-4}$ | $1.9e^{-3}_{\pm7e-4}$ | | $2.6e^{-3}_{\pm1e-3}$ | $2.7e^{-3}_{\pm1e-3}$ | $2.8e^{-3}_{\pm1e-3}$ | $1.0e^{-4}_{\pm4e-5}$ |

where $\rho$ is the density of the traffic, $f : \rho \mapsto \rho v(\rho)$ is the flux function and $v$ is the velocity. The flux function is typically modeled as a concave function of the density. Variations in the underlying velocity function give rise to different traffic flow models. In this work, six different models have been considered: Greenshields' (Greenshields et al., 1935), Triangular (Geroliminis and Daganzo, 2008), Triangular skewed (Geroliminis and Daganzo, 2008), Trapezoidal (Geroliminis and Sun, 2011), Greenberg (Greenberg, 1959) and Underwood (Underwood, 1961). These models behave *very* differently and should be considered as different equations, as shown in Figure 1. Formulations and illustrations of those six models are given in Appendix B.

**The inviscid Burgers' equation** is a well-known hyperbolic conservation law used in various domains such as fluid mechanics (Burgers, 1939), non-linear acoustics (Lombard et al., 2013), gas dynamics (Panayotounakos and Drikakis, 1995), and traffic flow (Musha and Higuchi, 1978). We refer the reader to Cameron (2011) for a thorough introduction. It is expressed as

$$\partial_t u + \frac{1}{2}\partial_x u^2 = 0. \tag{5}$$

This equation can be written in the classical form of a conservation law using the flux function $f : u \mapsto 1/2 \cdot u^2$. Exact solutions to Riemann initial conditions are also known for this problem. Visualization of some solutions, including videos, are available on our webpage and in Figure 1.

Our experiments focus on one-dimensional conservation laws in this work. Demonstrating that NFV can consistently outperform classical schemes in 1D establishes a strong foundation before tackling more complex systems. NFV architecture is, in principle, extendable to higher dimensions, since neural networks naturally scale to higher-dimensional inputs. Extending NFV to multi-dimensional

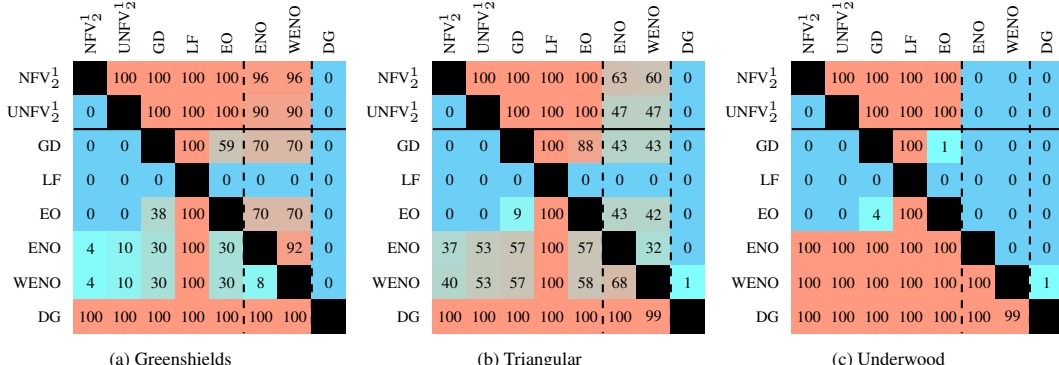

(a) Greenshields      (b) Triangular      (c) Underwood

Figure 3: **Comparison of numerical schemes across flow functions.** Each cell shows the proportion of the evaluation set on which the row scheme outperforms the column scheme. DG, the only FEM tested, is rarely beaten. NFV$_2^1$ and UNFV$_2^1$ outperform other first-order schemes and rival higher-order ones, making them strong choices depending on the equation.

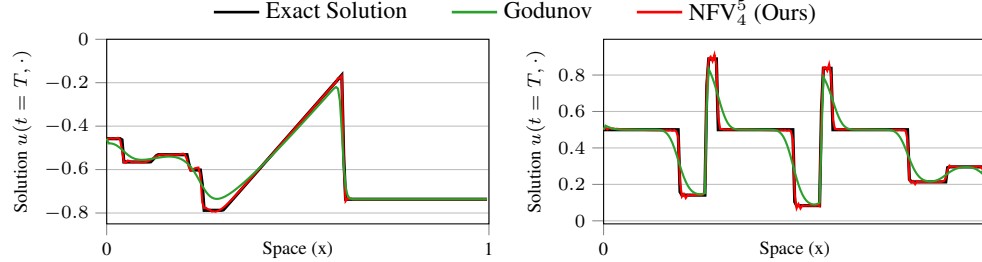

Figure 4: Comparison of the final density of the Burgers' equation (left) and LWR triangular equation (right) for $\text{NFV}_4^5$ and the Godunov Scheme. The proposed method displays an excellent approximation of the exact solution, capturing sharp features such as discontinuities and points of non-differentiability. It contains some minor oscillations in the solution, which are not present in the Godunov scheme. The latter, however, fails to capture the discontinuities and points of non-differentiability, offering a very smoothed solution.

Table 2: Evaluation of $\text{NFV}_4^5$ using piecewise constant initial conditions. Error is reported in $L_2$ norm. $\mathbf{NFV_4^5}$ achieves outstanding performance, gaining up to an order of magnitude improvement compared to Godunov and WENO. Its performance is close to DG, while keeping the implementation simplicity of a finite volume method and the computational complexity of NFV.

|              | Godunov            | WENO               | $\mathbf{NFV_2^1}$ | $\mathbf{NFV_4^5}$ | DG                 |
| ------------ | ------------------ | ------------------ | ------------------ | ------------------ | ------------------ |
| Burgers'     | $1.8e^{-3}_{\pm 6e-4}$ | $2.6e^{-3}_{\pm 1e-3}$ | $8.3e^{-4}_{\pm 3e-4}$ | $2.2e^{-4}_{\pm 1e-4}$ | $\mathbf{1.0e^{-4}_{\pm 4e-5}}$ |
| Greenshields | $4.1e^{-4}_{\pm 1e-4}$ | $6.9e^{-4}_{\pm 4e-4}$ | $1.2e^{-4}_{\pm 4e-5}$ | $4.6e^{-5}_{\pm 3e-5}$ | $\mathbf{4.2e^{-5}_{\pm 2e-5}}$ |
| Triangular   | $2.2e^{-3}_{\pm 1e-3}$ | $2.0e^{-3}_{\pm 2e-3}$ | $1.3e^{-3}_{\pm 6e-4}$ | $2.9e^{-4}_{\pm 2e-4}$ | $\mathbf{2.7e^{-4}_{\pm 1e-4}}$ |

will introduce additional challenges (e.g., numerical stability, computational complexity, and coupled variables), which we identify as important avenues for future work.

### 5.3 DATASETS

Training is performed using solutions derived from Riemann problems, which are initial value problems characterized by piecewise constant initial conditions with a single discontinuity (see Figure 8 for examples). These problems are fundamental in the study of hyperbolic PDEs and serve as essential test cases for numerical methods. For the scenarios considered in this work, analytical solutions to Riemann problems are available, making supervised learning possible. Evaluation is performed on a more complicated set of several hundred complex initial conditions to assess the model's generalization capabilities. These conditions consist of piecewise constant functions with ten discontinuities, giving rise to entropy solutions with multiple interacting shocks and rarefactions. Exact solutions for these test cases are computed using the Lax-Hopf algorithm (Lax, 1957; Claudel and Bayen, 2010a;b) on a finer grid (see Appendix D).

For the LWR benchmarks we train NFV autoregressively on 2048 randomly sampled Riemann problems with a single discontinuity, using discretization parameters $\Delta t = 5 \cdot 10^{-3}$, $\Delta x = 10^{-2}$, 100 spatial cells, and prediction horizons that are progressively increased from 10 to 250 steps under a robust CFL ratio of 0.5. Evaluation uses several hundred more complex piecewise-constant initial conditions whose exact solutions are computed on a finer grid with $\Delta t = 10^{-4}$, $\Delta x = 10^{-3}$, 200 cells, and 1000 time steps via the Lax–Hopf algorithm. Unsupervised UNFV models minimize a weak-form residual loss using 250 compactly supported polynomials of degree 50 as test functions over the spatial domain. Note that we use uniform space–time grids for simplicity and fair comparison to FV baselines, but the (U)NFV update depends only on cell volumes and interface fluxes and is therefore compatible with non-uniform or adaptive discretizations.

### 5.4 RESULTS AND DISCUSSION

Table 1 reports $L_2$ error for $\text{NFV}_2^1$, $\text{UNFV}_2^1$, and baseline methods across the seven benchmark equations. Our models consistently outperform all first-order FV methods, and surpass ENO/WENO schemes on about half of the equations. As expected, the higher-order DG method achieves signif-

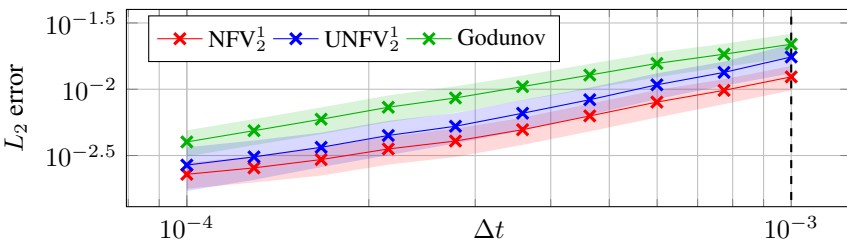

Figure 5: **Convergence plots on Greenshields' flux.** The $L_2$ error is computed against the exact solution on the evaluation set for different mesh discretizations. We report both error average and standard deviation, on a log-log scale. The dashed vertical line illustrates the discretization on which $NFV_2^1$ and $UNFV_2^1$ were trained; the models generalize to smaller discretizations. The ratio $\Delta t/\Delta x = 0.1$ remains constant as the mesh is refined.

icantly lower errors. Table 2 shows that $NFV_4^5$, while as simple to implement as standard $NFV_2^1$, achieves up to 10x better accuracy, approaching the performance of DG. In this sense, (U)NFV offers DG-level accuracy but with much smaller FV-like implementation complexity, substantially faster inference, and training that typically completes within fifteen minutes, while using memory comparable to Godunov and significantly lower than DG, since only the numerical flux is learned while the rest of the finite volume solver remains unchanged.

Figure 3 shows the fraction of test cases each method wins. $NFV_2^1$ and $UNFV_2^1$ consistently surpass first-order FV methods. Against ENO/WENO, performance varies: our models outperform on some equations, match on others, and underperform in a few, highlighting the complexity of benchmarking across diverse problem settings. Still, the fact that $NFV_2^1$ and $UNFV_2^1$ consistently do better than first-order methods is seen as a sign that the approach appears to converge well. Specifically, $NFV_2^1$ and $UNFV_2^1$ consistently produce errors bounded by those of Godunov, emphasizing their robustness.

Since all methods use autoregressive prediction, evaluating performance at the final time step provides a good proxy for cumulative error. Figure 4 shows that the prediction of $NFV_4^5$ closely aligns with the exact solution, with only minor oscillations observed. Notably, $NFV_4^5$ effectively captures sharp discontinuities with high accuracy without relying on smoothing techniques, which are commonly employed in traditional FV methods to mitigate numerical artifacts. Qualitatively, across the large set of complex test initial conditions that span weak to strong shocks and rarefaction patterns, NFV predictions remain visibly less diffusive than the others FV schemes, with sharp features dissipating more slowly while preserving stability.

**Ablation on discretization size:** Classical numerical schemes are known to converge as the grid is refined. Figure 5 shows that $NFV_2^1$ and $UNFV_2^1$ consistently achieve lower error than Godunov, a scheme proven to converge, across discretizations, suggesting that (U)NFV also converges to the entropy solution; the approximately linear trend in the log–log plot further indicates a polynomial convergence rate.

**Ablation on CFL ratio:** To further assess stability under different time step choices, we vary the CFL ratio while keeping the spatial grid fixed on Greenshields' LWR and report the resulting errors in Table 3. Across this range of CFL values, $NFV_2^1$ consistently attains lower mean error and substantially reduced variance compared to first-order FV baselines, and remains competitive with higher-order ENO and WENO schemes. The only exception is DG at very small CFL, which achieves the lowest error but becomes unstable and fails to run at higher CFL ratios, whereas $NFV_2^1$ remains robust.

Overall, the results support our hypothesis that training on simple Riemann problems is sufficient to generalize to complex piecewise-constant initial conditions. In particular, the ability of (U)NFV trained only on these analytically tractable Riemann building blocks to generalize reliably to much richer piecewise-constant and real-world configurations turns this seemingly strong assumption into a practical strength rather than a limitation. Additional dataset and training details, along with the heuristic exploration and hyperparameter tuning that led the method to work, are provided in Appendix D. In the next section, we show that NFV also generalizes to experimental highway data, where conservation is often violated and traditional methods typically fail.

Table 3: Mean and standard deviation of final-time $L_2$ error on the standard LWR benchmark with Greenshields' flux for different CFL ratios, comparing $\text{NFV}_2^1$ with classical finite volume baselines and DG.

| CFL | $\textbf{NFV}_2^1$ | GD | LF | EO | ENO | WENO | DG |
|---|---|---|---|---|---|---|---|
| 0.2 | $1.6e^{-4}_{\pm 3e^{-5}}$ | $3.8e^{-4}_{\pm 1e^{-4}}$ | $7.6e^{-3}_{\pm 2e^{-3}}$ | $3.8e^{-4}_{\pm 1e^{-4}}$ | $6.0e^{-4}_{\pm 4e^{-4}}$ | $6.2e^{-4}_{\pm 4e^{-4}}$ | $3.0e^{-5}_{\pm 1e^{-5}}$ |
| 0.4 | $1.3e^{-4}_{\pm 2e^{-5}}$ | $3.3e^{-4}_{\pm 1e^{-4}}$ | $4.1e^{-3}_{\pm 1e^{-3}}$ | $3.3e^{-4}_{\pm 1e^{-4}}$ | $6.0e^{-4}_{\pm 4e^{-4}}$ | $6.4e^{-4}_{\pm 4e^{-4}}$ | fail |
| 0.6 | $1.2e^{-4}_{\pm 5e^{-5}}$ | $2.1e^{-4}_{\pm 2e^{-4}}$ | $1.3e^{-3}_{\pm 4e^{-4}}$ | $2.2e^{-4}_{\pm 2e^{-4}}$ | $1.5e^{-2}_{\pm 1e^{-2}}$ | $1.5e^{-3}_{\pm 1e^{-3}}$ | fail |
| 0.8 | $1.0e^{-4}_{\pm 2e^{-5}}$ | $2.2e^{-4}_{\pm 7e^{-5}}$ | $2.0e^{-3}_{\pm 6e^{-4}}$ | $2.3e^{-4}_{\pm 7e^{-5}}$ | $1.6e^{-3}_{\pm 2e^{-3}}$ | $7.2e^{-4}_{\pm 4e^{-4}}$ | fail |
| 1.0 | $9.1e^{-5}_{\pm 2e^{-5}}$ | $1.7e^{-4}_{\pm 5e^{-5}}$ | $1.5e^{-4}_{\pm 5e^{-4}}$ | $1.8e^{-4}_{\pm 5e^{-5}}$ | $5.6e^{-3}_{\pm 6e^{-3}}$ | $9.6e^{-4}_{\pm 7e^{-4}}$ | fail |
| 1.2 | $1.2e^{-4}_{\pm 5e^{-5}}$ | $2.1e^{-4}_{\pm 2e^{-4}}$ | $1.3e^{-3}_{\pm 4e^{-4}}$ | $2.2e^{-4}_{\pm 2e^{-4}}$ | $1.5e^{-2}_{\pm 1e^{-2}}$ | $1.5e^{-3}_{\pm 1e^{-3}}$ | fail |

Table 4: **Improvements of NFV at different scales against numerical methods with fitted flow functions on field data.** The reported metrics include L1 error ($\text{mean}(|u - \hat{u}|)$), L2 error ($\text{mean}((u - \hat{u})^2)$), and relative error ($\text{mean}(|u - \hat{u}|/|\max\{\varepsilon, u\}|)$). The larger the input size of NFV, the better the performance. $\text{NFV}_2^1$ outperforms all calibrated Godunov fits, despite having the same input size and underlying structure.

| | Calibrated numerical schemes (Godunov) | | | | | NFV (Ours) | | |
|---|---|---|---|---|---|---|---|---|
| | Greenshields | Triangular | Trapezoidal | Greenberg | Underwood | $\text{NFV}_2^1$ | $\text{NFV}_4^5$ | $\text{NFV}_{10}^{11}$ |
| L1 | $6.05e^{-2}$ | $2.77e^{-2}$ | $2.73e^{-2}$ | $2.79e^{-2}$ | $4.98e^{-2}$ | $\mathbf{2.37e^{-2}}$ | $2.31e^{-2}$ | $\mathbf{2.02e^{-2}}$ |
| L2 | $1.93e^{-1}$ | $1.31e^{-1}$ | $1.30e^{-1}$ | $1.33e^{-1}$ | $1.81e^{-1}$ | $\mathbf{1.23e^{-1}}$ | $1.21e^{-1}$ | $\mathbf{1.09e^{-1}}$ |
| Rel. | $5.04e^{-1}$ | $3.83e^{-1}$ | $3.74e^{-1}$ | $3.75e^{-1}$ | $5.45e^{-1}$ | $\mathbf{3.57e^{-1}}$ | $3.51e^{-1}$ | $\mathbf{2.83e^{-1}}$ |

# 6 MODELING LARGE-SCALE EXPERIMENTAL FIELD DATA USING NFV

We apply the proposed NFV method to large-scale traffic field data collected on Interstate 24 (I-24) in Tennessee, USA, using the I-24 MOTION infrastructure (Gloudemans et al., 2023a;b), which enables high-resolution vehicle trajectory collection and constitutes the most extensive publicly available traffic dataset to date. Rather than predicting traffic speed, we focus on modeling traffic density, which is more directly tied to conservation laws and often exhibits sharp transitions that are challenging to capture. Although conservation of mass is not strictly satisfied in highway traffic data due to merges, exits, and incidents, it serves as a strong inductive bias. We show that NFV achieves superior predictive accuracy compared to classical numerical schemes. Moreover, incorporating the PDE structure leads to substantially more stable training, particularly in data-scarce regimes. These findings suggest that our approach can enhance the accuracy and efficiency of traffic simulations, thereby contributing to better-informed decision-making in urban planning and traffic management.

## 6.1 DATASET AND TRAINING

We evaluate our method on the I-24 MOTION dataset (Gloudemans et al., 2023a), which provides high-resolution vehicle trajectories collected on a four-mile stretch of Interstate 24 (mile markers 58.7 to 62.7) near Nashville, Tennessee. The data is captured by a network of high-definition cameras mounted along the highway as part of the I-24 MOTION infrastructure, leading to intricate trajectory data as illustrated in Figure 9. Vehicle trajectories are reconstructed using a computer vision and data association pipeline (Wang et al., 2022), resulting in high-fidelity, though inherently noisy, field data.

The dataset consists of 10 days of vehicle trajectory data, collected during the morning rush hour (6:00 AM to 10:00 AM) over the 4-mile segment. From the raw trajectory data, we construct spatiotemporal vehicle density fields by aggregating vehicle counts over fixed spatial cells. Details of the data cleaning, processing, and preparation are provided in Appendix C.1. Visualization of the resulting density fields is shown in Figure 10. Further training details are available in Appendix D.

Concretely, all NFV models and tuned finite-volume baselines are trained on the first hour of data from November 29, 2022 using a single boundary cell on each side; the autoregressive prediction horizon is increased from 10 to 100 steps during training, while the learning rate decays from $10^{-3}$ to $10^{-4}$ over roughly 3000–5000 epochs, leading to convergence within 15–30 minutes on a single NVIDIA RTX A5000 GPU.

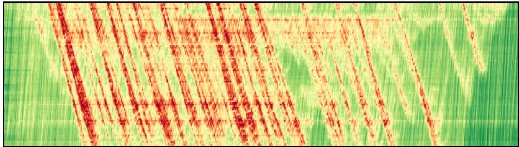 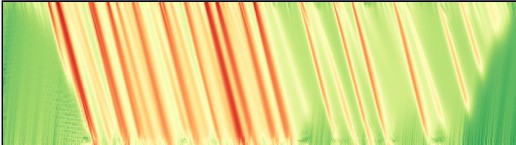

Figure 6: **Autoregressive prediction of NFV$_{10}^{11}$ (right) compared to the ground truth (left).** Full results are shown in Figure 13. See Appendix C.3 for how to read the heatmaps.

Table 5: **Generalization of NFV against Godunov on 7 days of I-24 data never seen during training.** As in Table 4, we report mean and standard deviation of L1, L2 and relative errors.

|  | L1 error | L2 error | Relative error |
| --- | --- | --- | --- |
| Godunov | $1.56\mathrm{e}^{-1} \pm 2.02\mathrm{e}^{-2}$ | $3.74\mathrm{e}^{-2} \pm 8.25\mathrm{e}^{-3}$ | $6.26\mathrm{e}^{-1} \pm 2.58\mathrm{e}^{-1}$ |
| NFV$_{10}^{11}$ | $\mathbf{1.12e^{-1} \pm 7.39e^{-3}}$ | $\mathbf{2.20e^{-2} \pm 2.59e^{-3}}$ | $\mathbf{3.59e^{-1} \pm 7.58e^{-2}}$ |

## 6.2 RESULTS AND DISCUSSION

We compare NFV to numerical schemes using the flux functions from Appendix B. These functions, each defined by a few parameters, were calibrated via optimization to minimize the Godunov scheme's prediction error on the training set. The search ranges were intentionally broad, prioritizing predictive performance over physical plausibility to ensure a fair comparison. We chose the Godunov scheme as the representative baseline to compare with since we observed only a marginal performance difference (up to 5%) between Godunov and other FV schemes on this dataset, and the Godunov scheme is known to converge to the entropy solution. We evaluate three NFV variants of increasing capacity: NFV$_2^1$, NFV$_4^5$, and NFV$_{10}^{11}$ (training details can be found in Appendix D), to assess how well they generalize and capture complex field dynamics.

Table 4 shows that all NFV models outperform the five tuned Godunov schemes, with performance improving as input size increases. This trend matches what was seen on synthetic data (Section 5). Despite training on just one hour of data, NFV predicts nearly four hours of traffic evolution autoregressively (Figure 6). While performance degrades in out-of-distribution zones (e.g., dark green regions unseen during training), the models still capture key wave patterns with high fidelity. Larger stencils help smooth out noise and improve accuracy, as seen in Figure 13.

We further evaluate generalization on 7 other days. As shown in Figure 14 and Table 5, NFV$_{10}^{11}$ consistently outperforms the best Godunov scheme on the evaluation set, even though both perform similarly on the training day. Indeed, although far from perfect, it is able to capture the evolution of free-flow traffic (dark green) with much greater accuracy, allowing it to successfully capture the end of congestion waves (red). NFV scales naturally with capacity: NFV$_{10}^{11}$ adds only 1728 parameters over NFV$_2^1$ but achieves significantly better accuracy with similar runtime and memory usage, unlike hand-crafted schemes, which significantly grow in complexity (see for example Appendix A).

## 7 CONCLUSION

We introduced (U)NFV, a neural network–based framework that extends finite volume methods for hyperbolic conservation laws by learning numerical fluxes over extended spatio-temporal stencils while preserving conservation. (U)NFV achieves high accuracy and efficiency, capturing complex wave dynamics with high fidelity, outperforming classical baselines on standard PDE benchmarks and large-scale field traffic data. Its modular design scales to large spatial and temporal stencils, matching the accuracy of state-of-the-art methods such as DG with much lower implementation complexity and significantly faster inference. In parallel work we have established convergence guarantees by controlling error propagation and deriving bounds on network size and training set requirements. These theoretical results will appear in a forthcoming journal publication. Future directions include applying (U)NFV to velocity-based formulations to learn speed-flux relationships, which aligns with GPS observations and avoids the need for a closed-form velocity PDE. Architecturally, NFV is dimension agnostic, but a systematic multi-dimensional empirical study remains for future work.

## REPRODUCIBILITY STATEMENT

To support reproducibility, we provide detailed descriptions of the NFV and UNFV architectures and training objectives in Sections 3 and 4, along with experimental setups in Sections 5 and 6. The formulations of all classical numerical baselines are presented in A, while benchmark equations are introduced in Section 5 and expanded in Appendix B. Additional implementation details, including model architecture, training procedures, hyperparameters, and dataset processing, are provided in Appendix D. Finally, code, datasets, benchmarks, and trained models are released at https://nathanlichtle.com/research/nfv.

## ETHICS STATEMENT

The submission does not have any ethics issues.

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

# A  FINITE VOLUME METHODS

Several finite volume-based numerical schemes are studied in this work. They include the following common classical first-order schemes:

**The Godunov method (Godunov, 1959a):**

$$\forall i, n \quad \hat{F}_{i-1/2}^n = \begin{cases} \min_{[u_{i-1}^n, u_i^n]} f & \text{if } u_{i-1}^n \leq u_i^n \\ \max_{[u_i^n, u_{i-1}^n]} f & \text{if } u_{i-1}^n > u_i^n \end{cases}$$

**The Lax-Friedrichs method (Lax, 1954):**

$$\forall i, n \quad \hat{F}_{i-1/2}^n = \frac{1}{2}\left(f(u_i^n) + f(u_{i-1}^n)\right) - \frac{1}{2}\frac{\Delta x}{\Delta t} \times |u_i^n - u_{i-1}^n|.$$

**The Engquist-Osher method (Engquist and Osher, 1981):**

$$\forall i, n \quad \hat{F}_{i-1/2}^n(u_{i-1}^n, u_i^n) = \frac{1}{2}\left(f(u_i^n) + f(u_{i-1}^n)\right) - \frac{1}{2}\int_{u_{i-1}^n}^{u_i^n} |f'|.$$

Additionally, higher-order schemes such as the **Essentially Non-Oscillatory (ENO) method (Shu, 1999)** and the **Weighted Essentially Non-Oscillatory (WENO) method (Liu et al., 1994)** are considered. The main idea in these methods is that by considering more stencils, one can expect to increase the accuracy of approximation of the solution.

For the ENO scheme, we consider the semi-discrete form of

$$\partial_t u_i = -\frac{1}{\Delta x}\left(\hat{F}_{i+1/2} - \hat{F}_{i-1/2}\right). \tag{6}$$

Using the Lax-Friedrichs Splitting technique, we have

$$f(u) = f^+(u) + f^-(u), \quad f^{\pm}(u) = \frac{1}{2}(f(u) \pm \alpha u), \tag{7}$$

where $\alpha = \max|f'(u)|$ is the maximum wave speed. The key point in the ENO scheme is the high-order upwind interpolation of $f^+$ and $f^-$ based on the smoothest stencils. For instance, for the 2-stencil ENO scheme, the procedure is as follows:

1. Evaluate the smoothness indicators:
$$\delta_- = |f_i^+ - f_{i-1}^+|, \quad \delta_+ = |f_{i+1}^+ - f_i^+|$$

2. Select the stencil that minimizes the smoothness indicator:
   - If $\delta_+ < \delta_-$, choose the stencil $\{f_i^+, f_{i+1}^+\}$.
   - Otherwise, choose the stencil $\{f_{i-1}^+, f_i^+\}$.

3. Perform linear interpolation to compute the numerical flux:
$$\hat{f}_{i+\frac{1}{2}}^+ = f_i^+ + \frac{1}{2}\delta^+$$

   where $\delta^+$ is the difference between the selected stencil elements.

A similar approach is applied to compute $\hat{f}_{i+\frac{1}{2}}^-$ using the right-biased stencil.

The final numerical flux at the interface is obtained by combining the positive and negative parts:

$$\hat{f}_{i+\frac{1}{2}} = \hat{f}_{i+\frac{1}{2}}^+ + \hat{f}_{i+\frac{1}{2}}^-$$

In this work, we have used a 3-stencil scheme for ENO.

The WENO scheme follows the same idea as ENO by using specific weights in defining $\hat{f}_{i+1/2}^+$, rather than explicit conditions. In this work, we use the 5-stencil WENO scheme.

# B  Variants of LWR

We consider six different LWR PDEs variants, each consisting of a different fundamental diagram, illustrated in Figure 7. All of the considered flows are concave continuous mappings from $[0, \rho_{\max}]$ to $\mathbb{R}_+$, where $\rho_{\max}$ is the maximum density, with the exception of the Greenberg flow whose domain is $(0, \rho_{\max}]$. The critical density $\rho_c$ denotes the density at which the flow is maximized, i.e. $\rho_c = \arg\max_{\rho \in [0, \rho_{\max}]} f(\rho)$. The following introduces the six flow models we consider in this work, each time detailing the flow's parameters, the parameter values we use in Section 5 (in parentheses), and the flow's definition. Note that we consider normalized parameter values lying between 0 and 1 for the most part.

**Greenshields**  Parameters: free-flow speed $v_{\max}$ (1 m/s), maximum density $\rho_{\max}$ (1 veh/m).

$$f(\rho) = v_{\max}\rho \left(1 - \frac{\rho}{\rho_{\max}}\right)$$

**Triangular 1 (symmetrical)**  Parameters: free-flow speed $v_{\max}$ (1 m/s), critical density $\rho_c$ (0.5 veh/m), maximum density $\rho_{\max}$ (1 veh/m), wave propagation speed ($-1$ m/s).

$$f(\rho) = \begin{cases} v_{\max}\rho & \text{if } \rho < \rho_c \\ w(\rho - \rho_{\max}) & \text{if } \rho \geq \rho_c \end{cases}$$

**Triangular 2 (skewed)**  A non-symmetric variant of the Triangular flow, with parameters $v_{\max} = 2$ m/s, $\rho_c = {}^1\!/_3$ veh/m, and $w = -1$ m/s.

**Trapezoidal**  Parameters: free-flow speed $v_{\max}$ (1 m/s), first density cusp $\rho_1$ (0.2 veh/m), second density cusp $\rho_2$ (0.8 veh/m), maximum density $\rho_{\max}$ (1 veh/m), wave propagation speed ($-1.5$ m/s).

$$f(\rho) = \begin{cases} v_{\max}\rho & \text{if } \rho < \rho_1 \\ (w(\rho_2 - \rho_{\max}) - v_{\max}\rho_1)\dfrac{\rho - \rho_1}{\rho_2 - \rho_1} + v_{\max}\rho_1 & \text{if } \rho_1 \leq \rho \leq \rho_2 \\ w(\rho - \rho_{\max}) & \text{if } \rho > \rho_2 \end{cases}$$

**Greenberg**  Parameters: maximum density $\rho_{\max}$ (1 veh/m), coefficient $c_0$ (2).

$$f(\rho) = c_0\rho \log\left(\rho_{\max}/\rho\right)$$

**Underwood**  Parameters: maximum density $\rho_{\max}$ (1 veh/m), coefficients $c_1$ (0.25) and $c_2$ (1).

$$f(\rho) = c_1\rho \exp\left(1 - c_2\rho\right)$$

Example solutions of the Greenshields LWR are shown on various initial conditions in Figure 8.

# C  I-24 Experimental Dataset: Density Extraction, Training, and Evaluation

## C.1  Density Fields Extraction From I-24 MOTION Dataset

I-24 MOTION is a large-scale traffic monitoring system installed along a section of Interstate 24 near Nashville, Tennessee. It uses a dense network of high-resolution cameras and computer vision algorithms to capture detailed, real-time vehicle trajectories across multiple lanes and miles of highway. The data collection network and resulting trajectory data are illustrated in Figure 9.

For our experiments, we use the INCEPTION dataset[1] (Gloudemans et al., 2023a) from I-24 MOTION, consisting of ten days of data, each covering the morning rush hour (6:00 AM to 10:00 AM). The dataset for each day comprises 15-20 GB stored as a single JSON file. We first split each file into

---

[1]Available at i24motion.org as part of the INCEPTION data release.

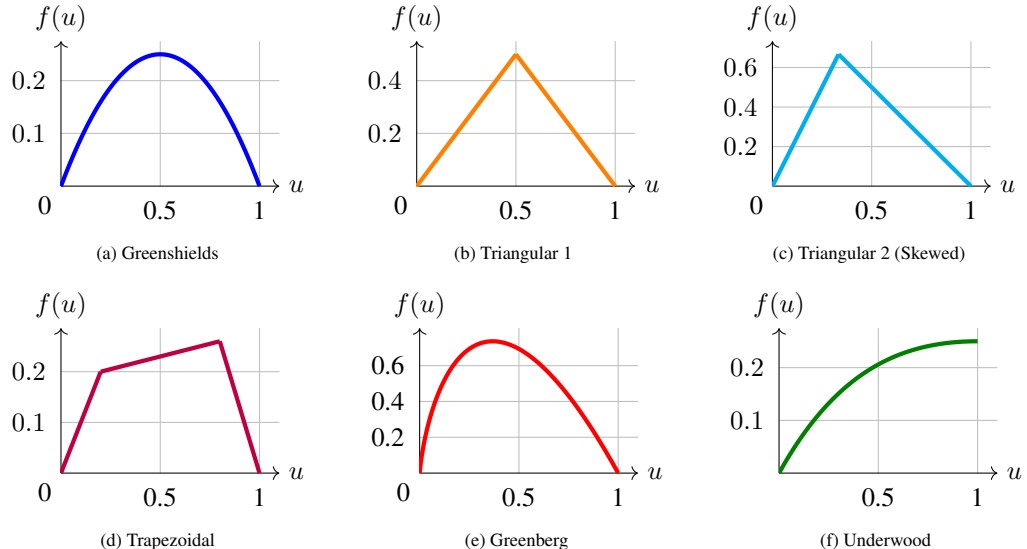

Figure 7: **Flow models for LWR.** We consider six different variants of the LWR PDE with the flows illustrated here, each mapping road density (veh/m) to traffic flow (veh/s).

manageable 1 GB chunks and parse them efficiently using `simdjson` (Langdale and Lemire, 2019), which enables extraction of density fields in approximately 3-5 minutes per 20 GB file.

To construct the density fields, we discretize the spatiotemporal domain into cells of size 0.02 miles ($\approx$32 meters) in space and 0.1 seconds in time, aggregating data across all four lanes. Vehicle counts in each cell are normalized to obtain densities in vehicles per kilometer per lane. To reduce noise, we average over 100 consecutive time steps (i.e., 10 seconds) and over 2 adjacent spatial cells (i.e., 0.04 miles or $\approx$64 meters). This results in a grid of 100 spatial cells (4 miles / 0.04 miles) and approximately 1440 time steps (4 hours / 10 seconds). We clip the first and last segments of each day to exclude low-density, free-flow regimes with incomplete data, retaining 1300 time steps per day depending on data quality. To avoid extreme outliers, we cap densities at 140 vehicles/km/lane. For all training and evaluation purposes, we then normalize densities so that the maximum density is 1.

Due to occasional sensor failures, such as malfunctioning camera poles or occlusions by bridges, there are seven spatial locations with missing data. We fill these gaps by linear interpolation between the adjacent upstream and downstream cells.

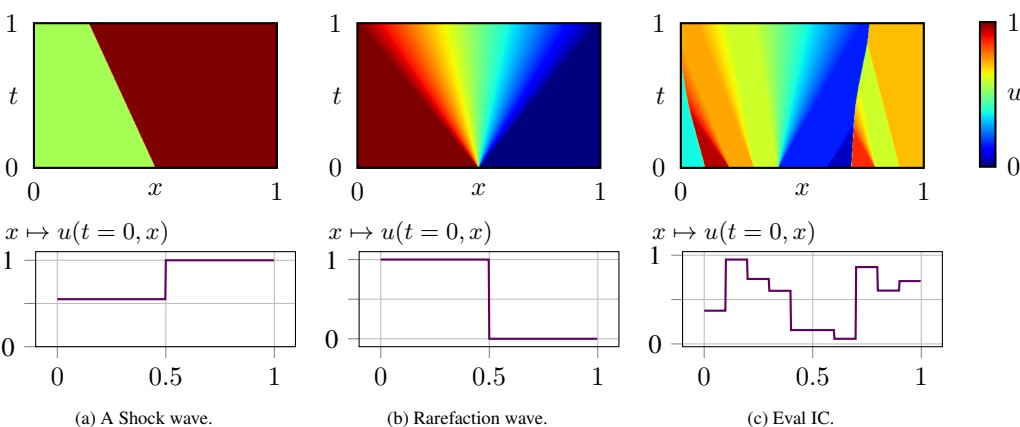

Figure 8: Exact solution for two Riemann problems (left, middle) and one piecewise-constant initial condition (right) from the evaluation set.

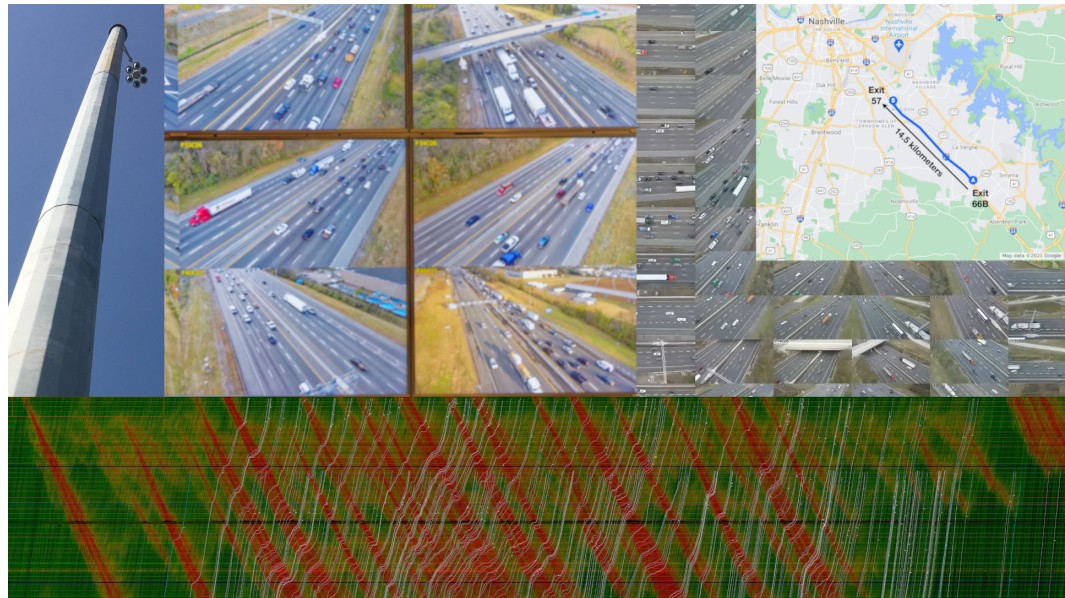

Figure 9: **I-24 MOTION illustration.** High-definition camera poles are mounted along a portion of I-24 at regular intervals. This generates massive amounts of video data, which is processed through a software stack. The resulting data for a single day is shown in the time-space diagram, displaying thousands of individual vehicle trajectories color-coded by speed (red for low speeds, green for high speeds), illustrating the complexity of the dataset.

Figure 10 shows the density fields we extracted from I-24 MOTION data. Higher densities (in red) correspond to stop-and-go waves and congestion, while lower densities (green) correspond to free-flow traffic. The processing code and resulting data are available in our codebase.

We exclude data from November 24 and 25, 2022, from our analysis, as both days correspond to holiday periods with purely free-flow, low-traffic conditions and no observable stop-and-go waves. These days are therefore not relevant to our study, which focuses on modeling traffic dynamics in the presence of congestion. The remaining days still include sufficient free-flow segments to evaluate model robustness in those regimes. Nevertheless, we include the excluded days in the released dataset for completeness.

## C.2 BOUNDARY CONDITIONS

For both training and evaluation of NFV on the I-24 dataset, we initialize the model using a single time step of real data and provide one boundary cell at each end of the road, using the corresponding real values. While it is possible to use additional ground truth data to improve accuracy, we deliberately restrict ourselves for two main reasons: (1) to allow fair comparison with the Godunov scheme, which uses a single boundary cell per side, and (2) to reflect realistic deployment scenarios, where boundary densities might only be measured at a few fixed points (e.g., at the road extremities), or predicted using a separate model.

For models that require a wider input stencil (e.g., those larger than $\text{NFV}_2^1$), we pad the boundaries by duplicating the available single-cell values. This ensures that all models, no matter their size: Godunov, $\text{NFV}_2^1$, or $\text{NFV}_{10}^{11}$, receive the same amount of boundary information. Figure 11 illustrates the boundary setup in both cases, showing which values are provided as input and which are left to be predicted.

Finally, we emphasize that initial and boundary conditions are not included when computing metrics, whether in the training loss or at evaluation.

### C.3 Reading the Heatmaps

This section provides a brief explanation and intuition for interpreting the heatmaps displaying I-24 MOTION data. The horizontal axis represents time, increasing from left to right, while the vertical axis represents space along the road, increasing from bottom to top. The color encodes traffic density, normalized between 0 and 1, according to the colormap shown in Figure 12, where green indicates low density traffic (free flow) and red indicates high density traffic (congestion). Unless otherwise specified, only the model predictions are shown, while initial and boundary conditions are omitted for clarity. Stop-and-go waves appear as high-density (red) bands that propagate upstream, i.e., move backward through traffic.

### C.4 Predictions

Predictions on the training day from Section 6.2 are displayed in Figure 13. Predictions on evaluation days are displayed in Figure 14.

## D Experiment Details

### D.1 Model Architecture

The model is applied locally on each cell to estimate the corresponding numerical flux. It is implemented as a two-dimensional CNN. The first layer uses a kernel of size $a$, followed by five convolutional layers with 15 channels and kernel size 1. Using a CNN enables efficient vectorized computation over all stencils, which is equivalent to sliding a fully connected network along the input but significantly faster. Each time step is represented as a separate input channel, for a total of $b$ input channels. Note that when $b = 1$, a one-dimensional CNN can be used. The output is a single channel providing the estimated flux. Each NFV model consists of 6 hidden layers of width 15, totaling $1105 + 16(ab + 1)$ parameters for $\text{NFV}_a^b$. This remains quite small, with around 1200 trainable parameters for the smallest variant, which is intentionally chosen as the smallest architecture that achieves competitive performance while consistently outperforming first-order FV baselines across our benchmarks.

We also found that activation functions have a modest effect: ELU activations perform slightly better on smooth flow functions (Greenshields, Greenberg, Underwood), while ReLU is preferable for piecewise-linear flows (Triangular, Trapezoidal). However, the difference in performance is minor.

### D.2 Training on Synthetic Data

For the LWR model, training is performed autoregressively: the model predicts future time steps by feeding its own outputs as inputs. We use 2048 randomly sampled Riemann problems $(\rho_1, \rho_2)$ for training, which proved more effective than uniformly spaced samples. To encourage generalization, small perturbations are added to the discontinuity location. Empirically, increasing the number of training Riemann problems beyond this scale did not yield noticeable accuracy gains and mainly increased training time, indicating that performance in this regime is not limited by data size.

The discretization parameters are $\Delta t = 5 \cdot 10^{-3}$, $\Delta x = 10^{-2}$, 100 space cells, and up to 250 time steps. Evaluation is done on 100 test cases generated with a finer grid and the Lax–Hopf algorithm: $\Delta t = 10^{-4}$, $\Delta x = 10^{-3}$, with 200 cells and 1000 time steps. A CFL ratio of 0.5 (e.g. $dx = 10^{-3}$ and $dt = 5 \cdot 10^{-4}$) was robust across different flow functions. Higher CFLs sometimes work but were less reliable. For instance, a CFL of 1.0 (e.g. $dx = dt$) is effective for the Greenshield flux but leads to poor performance on the Triangular flux.

The prediction horizon is progressively increased from 10 to 250 steps during training. Most progress occurs at short horizons (10 steps already outperform Godunov on average), while longer horizons provide additional fine-tuning and stability. The learning schedule that proved robust is summarized below:

For unsupervised experiments, we compute the weak loss function using test functions consisting of 250 randomly sampled compactly supported polynomials of degree 50. This proved to work well across PDEs, and details on test function generation are provided in the released codebase.

Empirically, we observed that performance is largely insensitive to the specific random draws, number, degree, or family of test functions once these values are moderately large, and that similar results are obtained with trigonometric test functions; degradation only appears for very small numbers or degrees (around 1-2), where the weak loss becomes poorly conditioned, so we found that the chosen setting offers a stable yet memory-efficient default.

| Stage | Training steps | Learning rate | $n_x$ | $n_t$ |
|-------|----------------|---------------|-------|-------|
| 1 | 10,000 | $1 \cdot 10^{-4}$ | 10 | 10 |
| 2 | 20,000 | $1 \cdot 10^{-5}$ | 50 | 50 |
| 3 | 20,000 | $5 \cdot 10^{-6}$ | 100 | 100 |
| 4 | 20,000 | $1 \cdot 10^{-6}$ | 200 | 200 |

Here, $(n_x, n_t)$ denotes the size of the space–time window predicted autoregressively. Most learning occurs during the first stage, with the later stages serving as progressive fine-tuning. Training uses the Adam optimizer with a decaying learning rate, from $10^{-4}$ to $10^{-6}$, and the largest batch size that fits in memory (ideally the entire dataset). Training on an RTX A5000 GPU takes about 30 minutes.

## D.3 TRAINING ON EXPERIMENTAL DATA

All models and fitted finite volume schemes are trained on the first hour of data from November 29, 2022, and evaluated on the full morning period (nearly four hours) and the remaining days of data. To ensure fairness and reflect practical deployment constraints, each model only receives a single boundary cell on each side, as described in Appendix C.2, even though larger models could benefit from additional context.

Each NFV model consists of 6 hidden layers of width 15, totaling $1105 + 16 \cdot (a \cdot b + 1)$ parameters for $\text{NFV}_a^b$. Training takes 15–30 minutes on an RTX A5000 GPU. The prediction horizon increases from 10 to 100 steps during training, while the learning rate decays from $10^{-3}$ to $10^{-4}$ over 3000–5000 epochs depending on model size, until convergence.

## D.4 HARDWARE AND RUNTIME

All experiments were run on a single NVIDIA A5000 GPU with 24GB of VRAM. The codebase (including (U)NFV and baselines) is fully vectorized so that all solutions are computed in parallel. Among classical schemes, Lax–Friedrichs is fastest due to its simplicity. Godunov, Engquist–Osher, and (U)NFV are 2–3× slower, ENO and WENO are 6× slower, and DG is up to 20× slower, with significantly larger memory requirements. Equivalently, NFV runs within a small constant factor of Godunov, is roughly twice as fast as ENO and WENO, and more than an order of magnitude faster than DG in our benchmarks, while still benefiting from GPU batching across hundreds of solutions where DG must be evaluated in smaller batches. These relative runtimes are hardware dependent but give a representative picture.

Training is also relatively fast on GPU. The model generally surpasses the Godunov baseline after only a few minutes and reaches most of its final performance within 15 minutes. In most runs we extended training to one hour or more, though the remaining time typically yields only minor fine-tuning. Figure 15 illustrates a typical training curve. To quantify progress, we also compute a *winrate* metric, defined as the percentage of evaluation initial conditions on which the model achieves lower $L_2$ error than a baseline such as Godunov. Under this metric, the model usually remains at 0% for the first few minutes, then rapidly increases to above 95%, and often reaches 100% winrate against Godunov within the first 15 minutes of training. A typical winrate curve is also depicted in Figure 15.

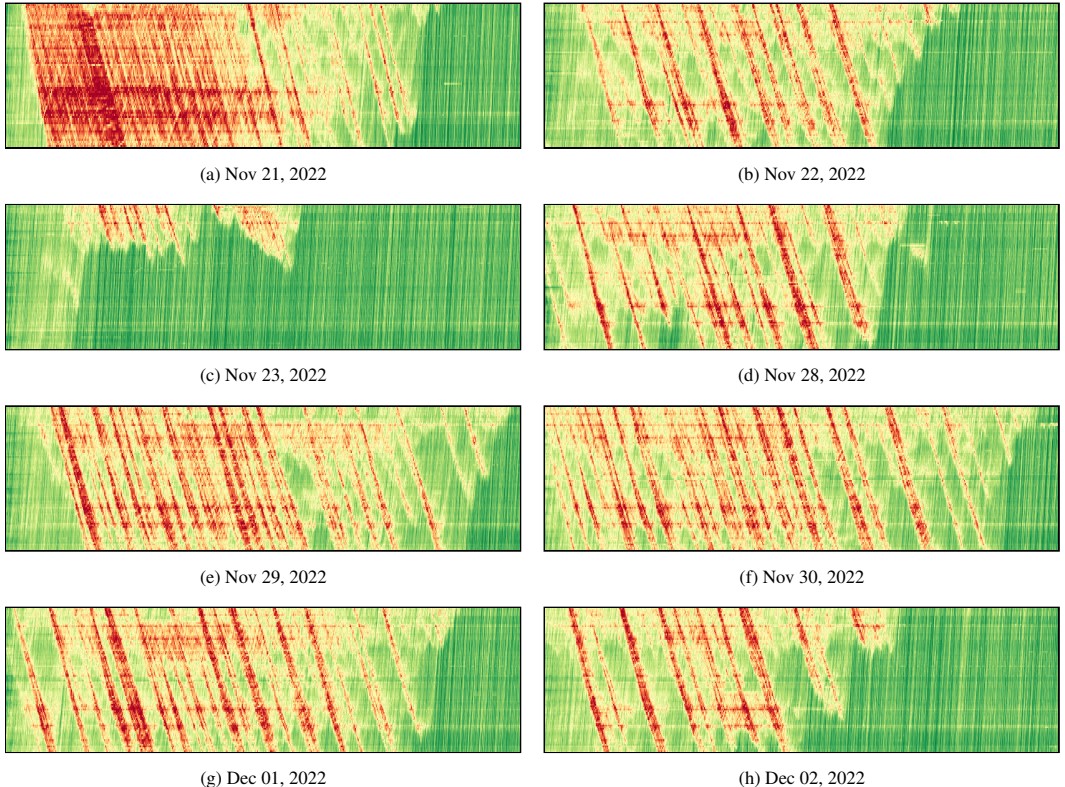

Figure 10: Time-space diagrams of car trajectories extracted from the video, colour-coded by density, for different dates. See Appendix C.3 for how to read the heatmaps.

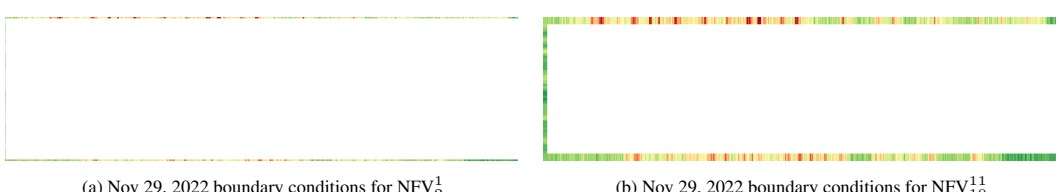

Figure 11: **Boundary conditions used by NFV during training and evaluation on the I-24 dataset.** The figures show the input provided to the model: the initial condition at $t = 0$ on the left, and boundary conditions at $x = 0$ (bottom) and $x = x_{\max}$ (top). The model must then predict the interior (i.e., the region shown in Figure 10) autoregressively: it uses its own output at time $t$ to predict the state at time $t + dt$, without receiving any additional data beyond the fixed boundaries. Note that both figures use the same underlying data; for $\text{NFV}_{10}^{11}$, the boundary values are duplicated to provide the required input padding.

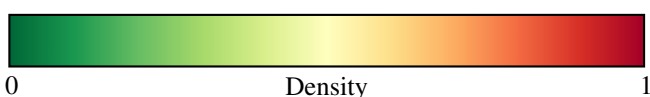

Figure 12: Colorbar showing density scale for all I-24 data heatmaps.

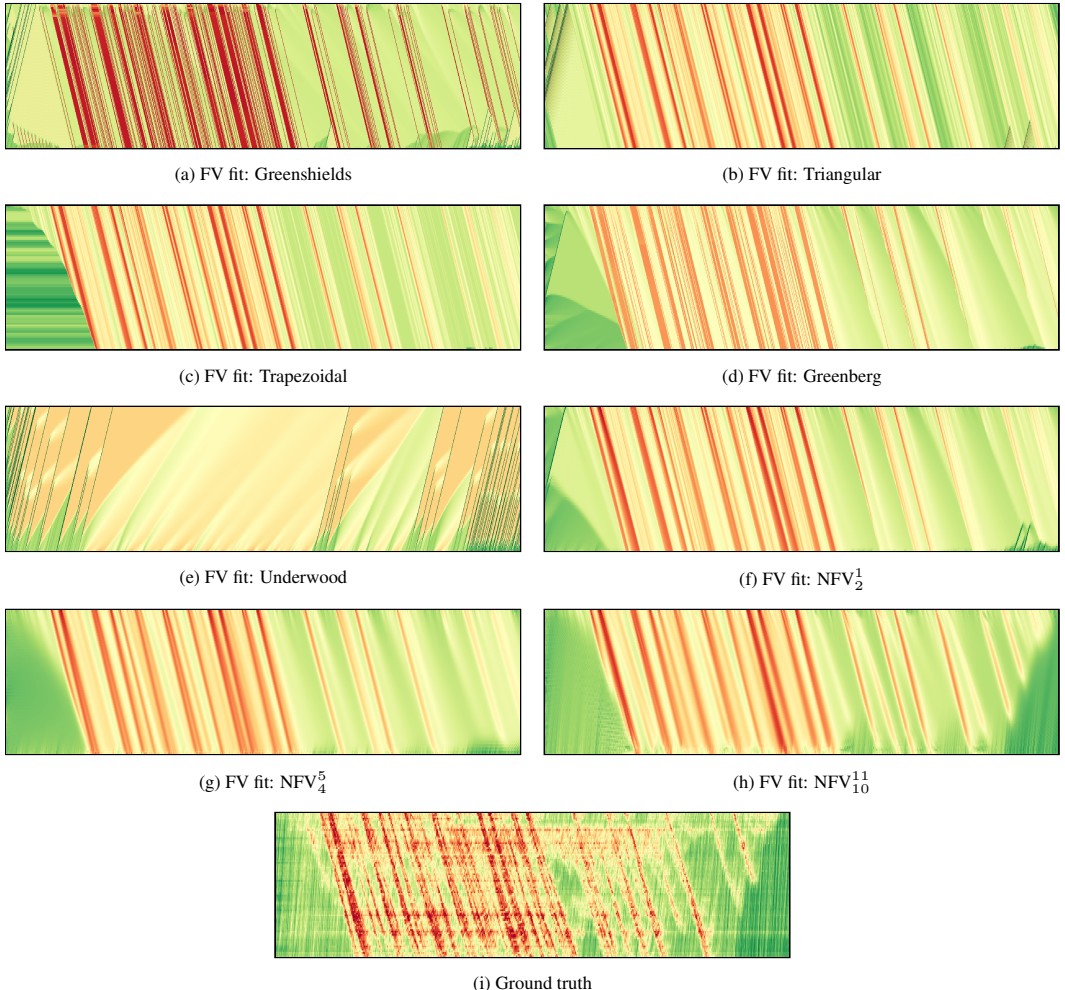

Figure 13: **Predictions of FV methods and trained NFV.** Corresponding metrics are reported in Table 4. Among the FV methods, only the Triangular, Trapezoidal, and Greenberg flows provide a reasonable fit to the I-24 MOTION data. In contrast, NFV models show increasing predictive accuracy with model complexity. For example, $NFV_{10}^{11}$ captures significantly more stop-and-go waves (in red) than $NFV_4^5$ or $NFV_2^1$, as well as fast low-density waves (in green), enabling it to correctly predict the early dissipation of the final two waves. However, it exhibits oscillations toward the end of the prediction window, likely due to limited generalization caused by the scarcity of low-density (dark green) patterns in the training data; nevertheless, the primary objective when modeling experimental data is to accurately capture the evolution of congestion waves, whereas free-flow traffic is of lesser interest. All models were trained on only the first 25% of the ground truth sequence, and the predictions are generated fully autoregressively. See Appendix C.3 for how to read the heatmaps.

**Prediction on training day**

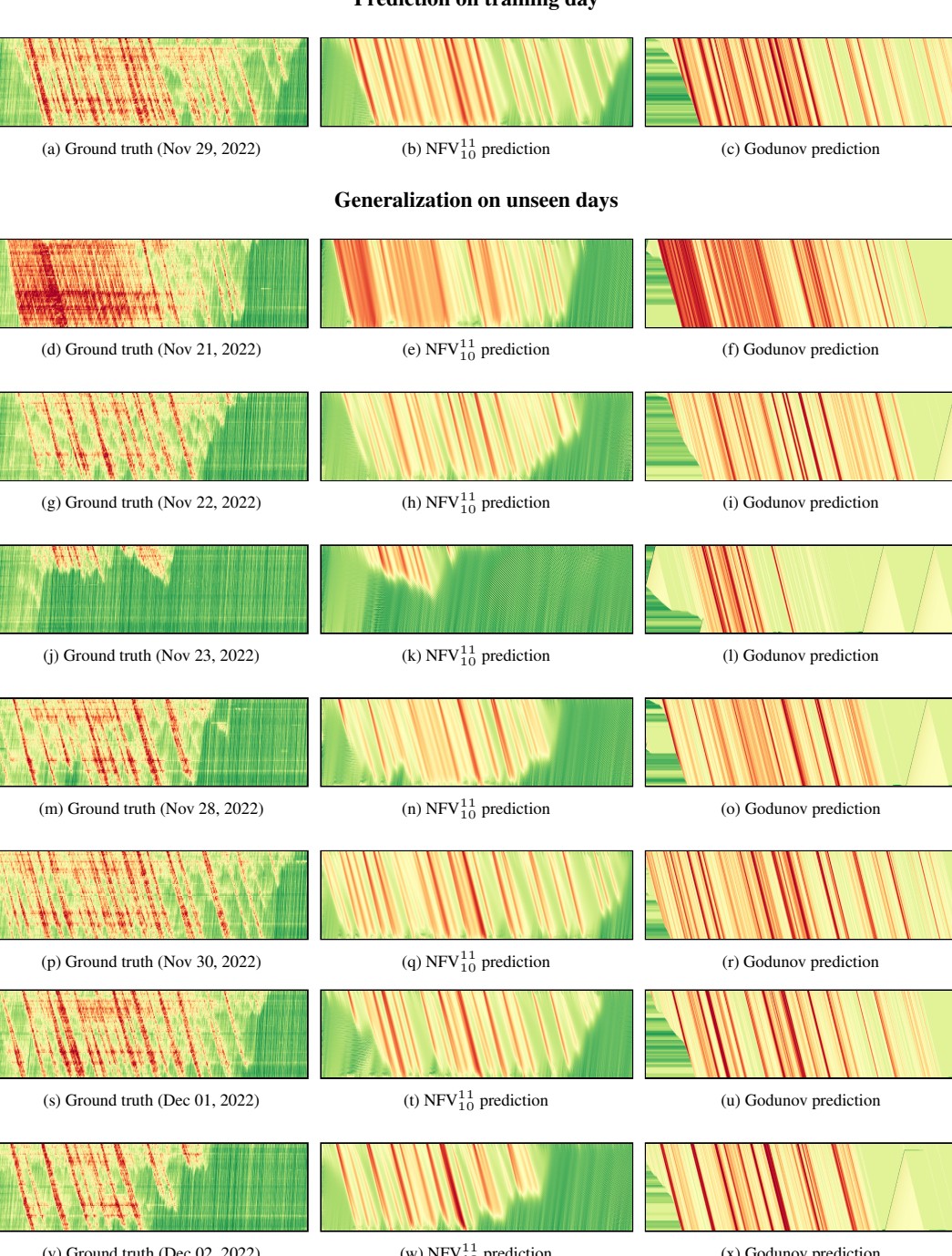

Figure 14: **Predictions of best FV fit and trained NFV$_{10}^{11}$.** Godunov is derived by fitting a flow function on the prediction and comparing it against the ground truth; we keep the fitted Trapezoidal flow as it performed best (see Figure 13 and Table 4). Both the Godunov fit and the NFV$_{10}^{11}$ training are realized using the same data, namely the first 1 hour of Nov 29, 2022 data (i.e., the first 25% of subfigure (a)). This means that the remainder of the data on that day (row 1), as well as the prediction on all subsequent days (rows 2-8) are generalization on data that was never seen before by either models. See Appendix C.3 for how to read the heatmaps.

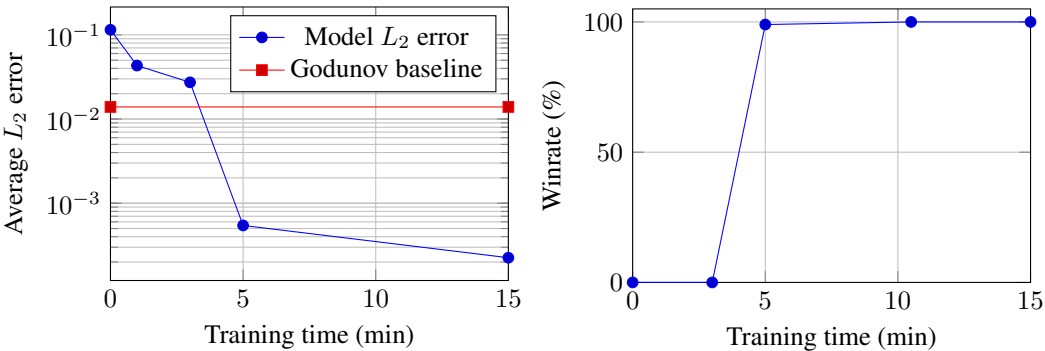

Figure 15: Training dynamics. **Left:** Average $L_2$ error of the model and of a Godunov baseline. **Right:** Winrate of the model against the Godunov baseline. All the metrics are computed periodically on an evaluation dataset of 100 random complex initial conditions, with prediction over 1000 timesteps.

