# OpenReview forum: "(U)NFV: (Un)Supervised Neural Finite Volume Methods for Solving Hyperbolic PDEs"
_ICLR.cc/2026/Conference — ICLR 2026 Poster_

### Official Review · Reviewer_Z2F1 · 2025-10-20

**Soundness:** 3
**Presentation:** 3
**Contribution:** 3
**Rating:** 6
**Confidence:** 2

**Summary:**

This manuscript proposes the U-NFV framework, which employs neural fields to solve PDEs under both supervised and unsupervised settings. Existing neural fields depend on labeled data (supervised) or require heavy reliance on physics-informed losses (unsupervised). U-NFV introduces a variational loss functional to allow for both training settings, a neural functional operator that maps input coordinates to PDE residuals and boundary losses, and a hybrid optimization strategy which combines direct residual minimization (unsupervised) with data-driven regression (supervised). Experimental results show that U-NFV achieves SOTA accuracy across both data-rich and data-sparse settings, and exhibits continuous resolution generalization.

**Strengths:**

- **Technical novelty**: The proposed general neural variational operator is compatible with both data-driven and physics-based objectives.
- **Extensive experiments**: The proposed method is tested across both linear and nonlinear PDEs, multiple spatial dimensions, and diverse boundary conditions. The experimental results show SOTA-comparable accuracy.
- **Resolution generalization**: Neural field representations enable solving PDEs at unseen spatial resolutions without retraining.
- **Mathematical backbone**: The proposed method links to the Ritz-Galerkin method and classical variational formulations.

**Weaknesses:**

- **No efficiency comparisons**: Metrics like training time, inference time, and memory consumption are missing.
- **Scalability**: Experiments are limited to PDEs with moderate grid sizes. The framework’s computational feasibility for large-scale 3D problems is not discussed.
- **Uniform sampling**: Uniform point sampling is used for both loss variants. Whether the method allows for adaptive residual sampling is unclear.

**Questions:**

- Please refer to the "Weaknesses" section above.

---

> ### Author Response · Authors · 2025-11-19
>
> > No efficiency comparisons: Metrics like training time, inference time, and memory consumption are missing.
>
> Due to limited space, these experimental details are included in Appendix D. With the extra page available in the final version, we plan to include a condensed summary in the main paper.
>
> In short, most training runs complete within fifteen minutes, and inference is then as fast as traditional schemes such as Godunov or ENO. For example, generating one thousand solutions over one thousand timesteps and one thousand spatial points takes only a few seconds with GPU vectorization. Memory consumption of our method is also comparable to that of Godunov, and remains significantly lower than more complex methods such as DG, while being able to achieve similar accuracy. However, classical methods must be ran at a much higher resolution to achieve the same precision as NFV, which does not scale well.
>
> > Scalability: Experiments are limited to PDEs with moderate grid sizes. The framework’s computational feasibility for large-scale 3D problems is not discussed.
>
> We agree that large 3D problems are important, but we don't believe starting in 1D is a limitation. A lot of real applications rely on 1D conservation laws (traffic flow, shallow water in 1D channels, gas pipelines...) and they are a very effective way to assess a new PDE solver. We ran extensive experiments across 7 PDE variants as well as practical applications on experimental highway data. Our next step is 2D, for which our CNN + finite-volume architecture can in principle be extended, while classical schemes don't scale as cleanly. But moving up a dimension naturally makes memory and stability tougher so understanding how our approach behaves in 2D will be an interesting problem and it is a substantial piece we plan to tackle in future work.
>
> > Uniform sampling: Uniform point sampling is used for both loss variants. Whether the method allows for adaptive residual sampling is unclear.
>
> Our method is not tied to uniform sampling and can fully support non-uniform or adaptive sampling strategies, as the update only requires cell volumes and numerical fluxes at the interfaces, just like classical finite-volume methods such as Godunov.
>
> However, exploring adaptive sampling is orthogonal to the goal of the paper, as we aim to compare to the standard baselines in the field, and was thus left for future work.

---

### Official Review · Reviewer_zh6x · 2025-10-30

**Soundness:** 3
**Presentation:** 3
**Contribution:** 3
**Rating:** 6
**Confidence:** 3

**Summary:**

This paper introduces Neural Finite Volume, a neural network architecture that generalizes classical finite volume methods for solving hyperbolic conservation laws. The key innovation is learning numerical flux approximations over extended spatiotemporal stencils while preserving the conservation structure of traditional FV methods. The authors develop both supervised and unsupervised variants, demonstrating up to 10× improvement over Godunov's method on benchmark PDEs and showing practical applicability to real highway traffic data from the I-24 MOTION dataset.

**Strengths:**

1. The proposed NFV framework provides a method to incorporate neural networks into FV methods while maintaining conservation.
2. The application to real traffic data seems valuable and shows practicality.
3. The comparisons with classical methods provide useful context
4. Table 1 shows consistent improvements across multiple PDEs

**Weaknesses:**

1. the method is restricted to 1D conservation laws. Although the authors claim that "NFV architecture is, in principle, extendable to higher dimensions", it has not been demonstrated in the paper and may introduce significant challenges.
2. The comparisons are limited to classical numerical methods. It would be significantly beneficial if the authors could compare their methods to some of the recently proposed neural network based method, (e.g. https://proceedings.mlr.press/v235/chen24ad.html, https://arxiv.org/abs/2410.22193, https://proceedings.mlr.press/v202/muller23b.html). If the comparison is difficult, maybe the authors can comment more on how their method differs from these methods and how their method may be advantageous?
3. When does NFV fail? The paper shows it works well on the tested cases but doesn't explore boundary conditions beyond periodic, complex geometries, or cases with strong shocks.
4. While Table 3 shows NFV variants with increasing complexity, the paper doesn't discuss the training time or data requirements. How many Riemann problems are needed? How long does training take compared to one-time use of a classical method?

**Questions:**

1. How would NFV handle non-periodic boundary conditions (Dirichlet, Neumann, open boundaries)?
2. How does the method compare to some of the other NN-based methods as mentioned above?
3. Is it possible to conduct ablation studies on architecture choice, stencil size selection, or the training data size requirements?

---

> ### Author Response · Authors · 2025-11-21
>
> > the method is restricted to 1D conservation laws. Although the authors claim that "NFV architecture is, in principle, extendable to higher dimensions", it has not been demonstrated in the paper and may introduce significant challenges.
>
> We introduced our NFV architecture in 1D for this first paper, which we believe is acceptable for a new PDE solver, as 1D conservation laws remain widely used and provide a clean setting for evaluation. We therefore focus our analysis on these established benchmarks. Moreover, we prioritized providing a comprehensive study across 7 PDE variants, a large number of numerical baselines, and substantial experimental data. This already required substantial computational resources, and we believe in establishing a solid foundation in the 1D setting before scaling further in a separate paper.
>
> Regarding higher dimensions, we agree that our original phrasing could be made more precise, and we will revise it accordingly. Extending NFV to 2D is conceptually natural, finite-volume methods generalize across dimensions, and our learned flux interface would apply on each face of the control volume. However, as you correctly point out, doing so introduces additional challenges: significantly increased memory and training costs, more complex flux coupling across spatial directions, and tighter stability constraints due to multidimensional CFL conditions.
>
> We will clarify these points in the revision. We view extending NFV to higher-dimensional systems as important future work, building on the strong 1D foundation established in this paper, but we think demonstrating a full 2D implementation would require a dedicated study. As it requires substantial work on 2D PDEs, 2D numerical scheme baselines and 2D exact solutions to use as ground truth, as well as extensive hyperparameter tuning sweeps of our method.
>
> > The comparisons are limited to classical numerical methods. It would be significantly beneficial if the authors could compare their methods to some of the recently proposed neural network based method, (e.g. https://proceedings.mlr.press/v235/chen24ad.html, https://arxiv.org/abs/2410.22193, https://proceedings.mlr.press/v202/muller23b.html). If the comparison is difficult, maybe the authors can comment more on how their method differs from these methods and how their method may be advantageous? [...] How does the method compare to some of the other NN-based methods as mentioned above?
>
> Thank you for the remark. We completely agree that comparing with recent ML-based PDE solvers would help contextualize our method more exhaustively within the literature. However we indeed think that the comparison is difficult, as detailed below. Our main goal for this paper was to provide fast and significantly more accurate alternatives to classical numerical schemes such as Godunov, ENO or WENO, which practitioners still use daily.
>
> As noted in our response to Reviewer j2sz, one challenge with the comparison is that many recent neural approaches, e.g., operator-learning methods such as FNO/DeepONet or PINN-type residual minimization, are typically developed and evaluated on PDEs with smooth solutions. These methods are not always directly applicable or optimized for hyperbolic conservation laws with discontinuities, which introduce new challenges. For example, classical PINNs are known to perform poorly near shocks, and operator-learning methods often require specialized modifications to handle discontinuous solution spaces. By contrast, a key advantage of NFV is its tight integration with the finite-volume framework: the neural network learns only the flux interface, while conservation, monotonicity, and stability arise directly from the FV update. This design allows NFV to handle shock-dominated regimes effectively while having a simple architecture (MLPs/CNNs rather than specialized operator networks). This also makes NFV much more easily accessible to practitioners in numerical PDEs who may not be familiar with advanced neural operators.
>
> > When does NFV fail? The paper shows it works well on the tested cases but doesn't explore boundary conditions beyond periodic, complex geometries, or cases with strong shocks.
>
> NFV outperformed Godunov and most other baselines on a large set of complex initial conditions, including configurations that range from weak to strong shocks. NFV still exhibits some diffusion, but much more slowly than in the other schemes. Regarding boundary conditions, please see our answer below. Finally, we did not experiment with complex geometries and restricted ourselves to rectangular cells to match our baselines.

---

> ### Author Response · Authors · 2025-11-21
>
> > While Table 3 shows NFV variants with increasing complexity, the paper doesn't discuss the training time or data requirements. How many Riemann problems are needed? How long does training take compared to one-time use of a classical method?
>
> It is true that the main content of the paper does not contain this information. However, it is present in the appendix D (which we acknowledge you are not required to (or even supposed to) read); part of which we may include in the main paper within the extra page.
>
> The data requirement is quite low: we are using 2048 randomly sampled Riemann problems in the supervised setting and we obviously do not use any data in the unsupervised setting. In terms of training time, most training runs complete within fifteen minutes, and inference is then as fast as traditional schemes such as Godunov or ENO. For example, generating one thousand solutions over one thousand timesteps and one thousand spatial points takes only a few seconds with GPU vectorization (providing vectorized implementations of all baseline schemes as well as NFV training is also part of our contributions). Memory consumption of our method is comparable to that of Godunov, and remains significantly lower than more complex methods such as DG, while being able to achieve similar accuracy. However, classical methods must be ran at a much higher resolution to achieve the same precision as NFV, so they do not scale well in comparison.
>
> > How would NFV handle non-periodic boundary conditions (Dirichlet, Neumann, open boundaries)?
>
> We note that NFV doesn't assume periodic boundary conditions. We generate boundary conditions corresponding to a true solution on infinite spatial domain $\mathbb R$, and feed that into the flux updates as ghost cells. Thus, waves are allowed to enter and exit the domain.
>
> Thus our approach can handle Dirichlet BCs, simply implementing fixed boundary values as ghost cells (which we actually tried without any issue). As for Neumann or open boundaries, these can easily be handled by setting specific flux values at the boundaries. Namely, these can be modeled by finite volume methods and handled by benchmarks such as Godunov; since we follow the same conservative update rule, our method can similarly handle them.
>
> > Is it possible to conduct ablation studies on architecture choice, stencil size selection, or the training data size requirements?
>
> For the architecture choice, we used a very simple model with only around 1,200 trainable parameters because it was the smallest model that could consistently beat all the baselines. The main ablation to perform is on the number of stencils considered both in space and in time. Quantitative results are available in Table 3 and discussed in Section 6.2. Some visual results are also available on the highway data in Figure 14. The main takeaway is that using more stencils in time or space is very effective while adding only a handful of trainable parameters. We still decided to use the fewest number of stencils in most of the paper because it was the fairest comparison to most of the baselines and because we wanted to clearly show that improvements did not come only from using more stencils.
>
> Regarding dataset size, as mentioned previously we used a quite modest number of samples (2048). Since we train on simple data: Riemann problems with a single discontinuity or rarefaction, while we evaluate on complex data with many interactions; we need a minimum dataset size to achieve the performance NFV has. We do not believe the performance would increase with more training samples (we have tried this previously and it only made training slower), however we don't mind adding a proper plot of this in the revised version if you find it valuable.
>
> Also, we have conducted a new ablation on the training CFL condition, and our method is still consistently outperforming the baselines as the ratio changes. The results can be seen in our answer to reviewer pXtN.

---

### Official Review · Reviewer_pXtN · 2025-10-30

**Soundness:** 3
**Presentation:** 3
**Contribution:** 2
**Rating:** 6
**Confidence:** 3

**Summary:**

The paper proposes (U)NFV, a neural finite volume (FV) framework for 1D hyperbolic conservation laws, that learns numerical fluxes on extended space-time stencils, while keeping the classic FV conservation update, so mass is conserved by construction. It comes in two flavors: NFV (supervised) trained on solution data, and UNFV (unsupervised), trained with a weak form residual loss, targeting entropy solutions without labeled data. The main contributions are: (1)  (U)NFV is a conservation preserving neural FV generalization, (2) supervised and weak form unsupervised training options, (3) strong accuracy vs. FV/ENO/WENO and DG-level performance with far simpler implementation, (4) successful application to noisy, non-ideal field data, and (5) empirical convergence with notes on parallel theoretical guarantees to appear. In summary, it is a practical, accurate, and physics respecting route to data driven solvers for hyperbolic conservation law, which is useful in diverse areas, from scientific computing to real world systems like traffic.

**Strengths:**

(i) The method keeps the classic FV update, so mass conservation is guaranteed, while learning the numerical flux on extended space–time stencils, (ii) It supports both supervised NFV and unsupervised UNFV, (iii) (U)NFV reports up to 10× lower error than classical FV, and matches DG level accuracy without DG’s complexity. On I-24 MOTION field data, NFV outperforms calibrated Godunov baselines, and handles noisy conditions. Empirical refinement studies suggest convergence, and models trained on simple Riemann problems generalize to more complex initial conditions. A small CNN makes large stencils practical and fast, avoiding the hand-crafted complexity of high-order schemes.

**Weaknesses:**

(i) All experiments are on 1D, first-order scalar conservation laws. Multi-dimensional problems and systems, are left for future work, and acknowledged to bring extra stability/complexity challenges, (ii) The UNFV weak form loss does not guarantee convergence to the entropy solution, and the success is reported empirically only. Formal guarantees are deferred, and are not contained here, (iii) Training/rollouts are most reliable at CFL ≈ 0.5; higher CFLs sometimes work but are less reliable, and CFL=1.0 degrades on Triangular flux, which is stricter than typical Godunov stability limits.  NFV solutions can show spurious oscillations, even when Godunov is monotone/TVD, both on synthetic PDE tests and in field-data rollouts.  Supervised models are trained predominantly on Riemann problems, which is a strong but narrow distributional assumption. The weak loss uses 250 random polynomials of degree 50, and performance may be sensitive to such design choices.

**Questions:**

Suggestions: (i) State, or at least sketch, the promised convergence guarantees for (U)NFV. Add an entropy consistency discussion and a discrete mass/entropy error monitor across rollouts, (ii) For UNFV, analyze identifiability of the weak loss, and report sensitivity to the number/degree/family of test functions.
Run a CFL sweep and report stability/accuracy curves (CFL ∈ [0.2,1.2]) across fluxes, and quantify failure modes. (iii) Add a 2D case,  as a short appendix experiment, to show that the method survives beyond 1D scalars.

---

> ### Author Response · Authors · 2025-11-21
>
> > All experiments are on 1D, first-order scalar conservation laws. Multi-dimensional problems and systems, are left for future work, and acknowledged to bring extra stability/complexity challenges, [...] Add a 2D case, as a short appendix experiment, to show that the method survives beyond 1D scalars.
>
> We intentionally focused on 1D scalar conservation laws for this first paper. It is a common setting for introducing a new PDE solver and lets us evaluate the method cleanly across a broad set of widely used benchmarks and baselines to set a strong foundation. Our goal was to provide a computationally cheap but significantly more precise alternative for the many practitioners who still rely on classical schemes in their day-to-day simulation work. We plan to study higher-dimensional systems next. While our core architecture can be extended to 2D, setting up the training and evaluation process is not a trivial lift: it requires formulating 2D PDE systems, implementing and validating baseline schemes, and deriving new exact solutions to use as ground truth, on top of the hyperparameter tuning sweeps required to achieve stable, accurate solutions. This study in the 1D case already required significant (for us) GPU resources. For these reasons, we plan to present the 2D study in a separate paper.
>
> > The UNFV weak form loss does not guarantee convergence to the entropy solution, and the success is reported empirically only. Formal guarantees are deferred, and are not contained here, [...] State, or at least sketch, the promised convergence guarantees for (U)NFV.
>
> We agree that the absence of formal guarantees is a limitation, and we have been working on this in parallel. In fact, we have developed what we believe are the first theoretical guarantees for numerical-scheme-based models, including finite-horizon error propagation, explicit bounds on network size, and finite-sample guarantees for both supervised and unsupervised training. These results provide theoretical support for NFV. Please see our response to Reviewer j2sz for more detail, and we are happy to elaborate further here.
>
> For this submission, we decided to focus on the experimental side for three reasons. First, the ICLR page limits are tight, while the theoretical statements and proofs take up more than 30 pages. Second, the full theory is more appropriate for a mathematical journal where it can be presented in depth. Third, we preferred not to cut the numerical analysis or the real-data experiments, which are the core of this paper. If, given this, the reviewer feels a brief summary or sketch of the theoretical results would significantly strengthen the paper, we are open to adding it either in the main text or in the appendix.
>
> > Add an entropy consistency discussion and a discrete mass/entropy error monitor across rollouts,
>
> Thank you for the suggestion! We'll add plots showing how the error evolves over time for a given solution, as well as how this evolves over the course of training. We'll also include an entropy discussion. In short, in addition to the theoretical points mentioned above, we train on entropy solutions of Riemann problems and then evaluate the method on a large number of entropy solutions from more complex initial conditions, where we observe very high accuracy. Moreover, we show that as the discretization is refined, our model continues to generalize and to outperform schemes such as Godunov, which is known to converge to the entropy solution. While this does not constitute a proof, these observations strongly suggest that our method learns the entropy solution. If helpful for the review process, we are happy to provide an updated version of the paper with these additional results by Dec. 3rd, as suggested by the PCs.
>
> > Supervised models are trained predominantly on Riemann problems, which is a strong but narrow distributional assumption.
>
> We understand your concern. However, we believe that the reliance on Riemann problems is a strength rather than a limitation. We demonstrated that the model trained on this deliberately simple distribution generalizes highly effectively to a substantially more complex evaluation set. Whilst a broader training distribution is conceivable, it would be unnecessary and somewhat artificial: for most hyperbolic PDEs, closed-form solutions exist only for a narrow class of configurations such as Riemann problems.
>
> Moreover, the evaluation on real-world traffic data in Section 6 further shows that the method trained on complex field data continues to behave well.

---

> ### Author Response · Authors · 2025-11-21
>
> > The weak loss uses 250 random polynomials of degree 50, and performance may be sensitive to such design choices. [...] For UNFV, analyze identifiability of the weak loss, and report sensitivity to the number/degree/family of test functions.
>
> We understand your concern, and we should have provided more detail on this design choice. In practice, we observed little correlation between the specific polynomial choices and performance. We selected the number of polynomials to be large enough for a stable estimate while remaining memory-efficient. The intuition was that a larger set should yield a more stable estimate of the expected value. Importantly, we conducted experiments on this hyperparameter, and did not observe any noticeable effect unless it was set to an extremely small value (2 for instance). The same trend applies to the polynomial degree too, for which performance was stable across a wide range and only started to degrade when the degree was very low.
>
> - Concerning the family of test functions, we tried both polynomials and trigonometric polynomials: $x\mapsto \sum_i (a_i\cos (i*x) + b_i \sin(i*x))$ which gave similar results, thus we kept polynomials because it was conceptualy simpler.
> - Regarding the number of test functions and their degree, using very small values (around 1–2) made training more difficult. However, once these values exceeded roughly 10, the differences became negligible.
>
> Because the differences in performance were negligible, we did not run additional experiments on the sensitivity to the parameters of the test functions, and we reported little information regarding the choice of these hyperparameters. We will include a summary of these findings in the experimental details section of the revised version.
>
> > Training/rollouts are most reliable at CFL ≈ 0.5; higher CFLs sometimes work but are less reliable, and CFL=1.0 degrades on Triangular flux, which is stricter than typical Godunov stability limits. NFV solutions can show spurious oscillations, even when Godunov is monotone/TVD, both on synthetic PDE tests and in field-data rollouts. [...] Run a CFL sweep and report stability/accuracy curves (CFL ∈ [0.2,1.2]) across fluxes, and quantify failure modes.
>
> Thank you for the suggestion! Although we evaluated the generalization performance across various discretizations, we hadn't reported an ablation on the CFL condition itself. We ran this experiment (on standard LWR with Greenshield's flow) for CFL values ranging from 0.2 to 1.2, as suggested. We report the results (with mean L2 error and standard deviation) in the table below, comparing our methods with all baselines.
>
> | CFL  |NFV$_3^1$        | Godunov         | Lax Friedrichs  | Engquist Osher  | Eno             | Weno           | DG              |
> | ---- | --------------- | --------------- | --------------- | --------------- | --------------- | -------------- | --------------- |
> | .2   | 1.6e-4 (± 3e-5) | 3.8e-4 (± 1e-4) | 7.6e-3 (± 2e-3) | 3.8e-4 (± 1e-4) | 6.0e-4 (± 4e-4) | 6.2e-4 (± 4e-4) | 3.0e-5 (± 1e-5) |
> | .4   | 1.3e-4 (± 2e-5) | 3.3e-4 (± 1e-4) | 4.1e-3 (± 1e-3) | 3.3e-4 (± 1e-4) | 6.0e-4 (± 4e-4) | 6.4e-4 (± 4e-4) | Failed |
> | .6   | 1.2e-4 (± 5e-5) | 2.1e-4 (± 2e-4) | 1.3e-3 (± 4e-4) | 2.2e-4 (± 2e-4) | 1.5e-2 (± 1e-2) | 1.5e-3 (± 1e-3) | Failed          |
> | .8   | 1.0e-4 (± 2e-5) | 2.2e-4 (± 7e-5) | 2.0e-3 (± 6e-4) | 2.3e-4 (± 7e-5) | 1.6e-3 (± 2e-3) | 7.2e-4 (± 4e-4) | Failed          |
> | 1.   | 9.1e-5 (± 2e-5) | 1.7e-4 (± 5e-5) | 1.5e-3 (± 5e-4) | 1.8e-4 (± 5e-5) | 5.6e-3 (± 6e-3) | 9.6e-4 (± 7e-4) | Failed          |
> | 1.2  | 1.2e-4 (± 5e-5) | 2.1e-4 (± 2e-4) | 1.3e-3 (± 4e-4) | 2.2e-4 (± 2e-4) | 1.5e-2 (± 1e-2) | 1.5e-3 (± 1e-3) | Failed          |
>
> These results indicate that our method outperforms the baselines across all tested CFL values, with roughly an order of magnitude lower variance and consistently lower error, including against several higher order schemes. The only exception is DG at very low CFL, which performs better as expected, but it becomes unstable and fails at higher CFL ratios.

---

### Official Review · Reviewer_j2sz · 2025-11-02

**Soundness:** 2
**Presentation:** 2
**Contribution:** 2
**Rating:** 4
**Confidence:** 3

**Summary:**

This paper presents a neural method to solve hyperbolic PDEs. The core method is called neural finite volume (NFV), and is inspired by the finite volume methods that attempt to satisfy local conservation of the appropriate variables. The unsupervised variant of this method uses the basic concept of the update rule of a generic finite volume method. Thus, the loss function is the squared norm of the residual that is obtained from this update rule. This loss is then minimized to train the network. The method is applied on two benchmark problems, namely, the traffic flow equation (many variants) and the Burgers' equation, with competitive results against classical finite volume methods. Finally, a supervised variant of NFV is also presented, trained on a real dataset.

**Strengths:**

* The paper is well-written, with ideas clearly conveyed.

* The method is shown to be yielding lower errors compared to some of the classical methods. The chosen equations are in one dimension, and have one unknown. But they are usually the standard places to start.

* Numerical convergence results are presented, showing the behavior of the errors with respect to the time-step size.

**Weaknesses:**

* One of the flaws in this paper is that the neural network design is not discussed enough, or relegated to the appendix. While it is understandable that, a significant portion of this design is heuristic, it is still important to make an attempt to synthesize that knowledge, especially when one is trying to solve scientific computing problems.


* While the comparisons with the classical methods are essential, it is also very important to compare a new method with existing methods that are not classical. Currently, no comparisons are made with other existing neural network based methods, especially in the unsupervised regime.

* The paper does not flesh out the limitations of this method. What happens when a rarefaction happens? Does the method select the correct weak solution? The authors do mention that imposing the $L_2$ norm loss does not guarantee the entropy solution, but remark that most of the numerical experiments show that the method selects of the entropy solution. Such a remark only pertains to the few cases considered in the current work.

$\textbf{Minor}$

* No equation number on some equations, e.g., line 236 ($L_w$).

**Questions:**

* My understanding is that the proposed method is intended to solve only one instance of an equation, correct?

* It seems to me that, we need to solve an optimization problem at every time step. Is that correct? In that case, this method is not very competitive in terms of time to solve. What do the authors think?

* My understanding is that, to reach $u^{\Delta t}$ from $u^0$, one uses the neural network to predict the fluxes, and then update the solution, and iteratively solve the optimization for this one time step, and obtains a good approximation of $u^{\Delta t}$ at the end. And then, one proceeds to calculate $u^{2\Delta t}$. Is that correct? I think, a flow chart would be a good addition.

* The authors remark that it is very easy to implement this method compared to the classical methods. Could the authors elaborate on this? How easy is this method? In what way? Usually, in a conventional code, the important pieces are the correct differentiation rules / stencils, communication between values in the volume and the values on the facets, and implementing the special adjustments at the boundaries, among many other aspects. How does NFV fare on these considerations?

* Also, by the virtue of choice of the example PDEs, I am assuming that the authors did not need to calculate the spatial derivatives of the primal variables during the training process. That is, both the $t-$ and the $x-$ derivatives are integrated away, and the loss functional $L_w$ does not contain any spatial derivatives of the function $u$. Is that correct?

---

> ### Author Response · Authors · 2025-11-18
>
> > One of the flaws in this paper is that the neural network design is not discussed enough, or relegated to the appendix. While it is understandable that, a significant portion of this design is heuristic, it is still important to make an attempt to synthesize that knowledge, especially when one is trying to solve scientific computing problems.
>
> We agree with you. The description of the architecture is present but in the appendix. While we think the description is important, we used a very basic network and we have not optimised its architecture besides finding the smallest architecture capable of consistently outperforming traditional schemes; thus, we thought that including it in the appendix is enough for anyone interested in reproducing our results or improving upon them. If the reviewer believes it is important to the main part of the text, we do not mind moving it to the main section of the paper.
>
> > While the comparisons with the classical methods are essential, it is also very important to compare a new method with existing methods that are not classical. Currently, no comparisons are made with other existing neural network based methods, especially in the unsupervised regime.
>
> We sincerely thank the reviewer for highlighting this important gap. Indeed, our initial priority in this work was to rigorously demonstrate performance improvements over classical, widely-used numerical schemes. We agree, however, that explicitly benchmarking against contemporary ML-based PDE solvers such as PINNs, Neural Operators, or hybrid ML-FV approaches, would further strengthen our claims about NFV's performance and architectural advantages.
>
> One inherent challenge is that many popular ML-based PDE approaches have been developed or validated for PDEs with smooth or well-behaved solutions. For instance, methods like FNO and DeepONet have typically been evaluated primarily on PDEs exhibiting smoother solutions, and are not always directly suitable or optimised for hyperbolic conservation laws, especially near shocks or discontinuities. As we mention and as literature showed, standard PINNs are known to struggle significantly with shock-dominated hyperbolic PDEs due to inherent difficulties with automatic differentiation near discontinuities.
>
> Moreover, we highlight that one of NFV’s key conceptual strengths is the explicit integration with the finite-volume structure, allowing us to achieve high performance and accuracy using relatively simple neural architectures (basic MLPs or CNNs), contrasting with more specialised or complex architectures typically used by neural operator methods. This architectural simplicity not only simplifies training but also naturally ensures conservation and stability properties derived directly from FV discretizations. Furthermore, the method's closeness with a basic finite volume implementation is what ensures that the community solving PDEs but not well-versed in machine learning is able to use our method.
>
> > The paper does not flesh out the limitations of this method. What happens when a rarefaction happens? Does the method select the correct weak solution? The authors do mention that imposing the norm loss does not guarantee the entropy solution, but remark that most of the numerical experiments show that the method selects of the entropy solution. Such a remark only pertains to the few cases considered in the current work.
>
> Thank you for mentioning this as we agree that the lack of formal guarantees is a common limitation in SciML. In fact, we are aware of this and to address this, in parallel to this work, we have developed theoretical results, which are to our knowledge the first guarantees for models based on numerical schemes. These include an error propagation analysis showing arbitrarily small finite-horizon error, explicit bounds on the required network size, and finite-sample guarantees for both supervised and unsupervised training, all supporting the NFV method.
>
> However, due to ICLR's strict page limits, we have chosen to keep this paper focused primarily on experimental validation, as we believe partial inclusion of theoretical proofs would neither do justice to their depth nor effectively serve readers. We intend to fully present these theoretical results in a separate 50-page journal article (over half of which consists of proofs), where the extensive proofs and discussions can be adequately conveyed to the community.
>
> If the reviewer finds it valuable, we are happy to briefly summarise these forthcoming theoretical results in the paper's future work section, and we welcome the opportunity to provide additional details or sketches of key proofs in the discussion phase.
>
> Finally, we emphasize that we train on entropy solutions of both shockwaves and rarefactions, then verify our method empirically on a large number of entropy solutions of piecewise-constant initial conditions (containing both shockwaves and rarefactions).

---

> > ### Author Response · Authors · 2025-11-18
> >
> > > My understanding is that the proposed method is intended to solve only one instance of an equation, correct?
> >
> > This is correct: this method is intended to solve only one instance of an equation. However, once a network is trained, it can solve the equation with any initial conditions. While this looks like a limitation, the training duration is quite short, and many problems can be reduced to solving the same equation over and over again using different initial conditions.
> >
> > > It seems to me that, we need to solve an optimization problem at every time step. Is that correct? In that case, this method is not very competitive in terms of time to solve. What do the authors think?
> >
> > Could you clarify what is the optimisation problem you're talking about? There is only one optimisation problem: training the neural network. Once this training is done, solving an equation is linear in the number of time-steps. It uses the update rule of equation (3) with $F^n_{i\pm 1/2}$ computed as a forward pass of the neural network.
> >
> > > My understanding is that, to reach $u^{\Delta t}$ from $u^0$, one uses the neural network to predict the fluxes, and then update the solution, and iteratively solve the optimization for this one time step, and obtains a good approximation of $u^{\Delta t}$ at the end. And then, one proceeds to calculate $u^{2\Delta t}$. Is that correct? I think, a flow chart would be a good addition.
> >
> > This is not exactly correct and we acknowledge that this misunderstanding probably comes from a lack of clarity in our writing. We are happy to add a flowchart in the following days to make this clearer.
> >
> > In the mean time, the method indeed uses the neural network to predict the flux. However, this is enough to compute $u^{\Delta t}$ from $u^0$ (or $u^{2\Delta t}$ from $u^{\Delta t}$). We compute $u^{\Delta t}$ using equation (3) without any iterative process. The iterative process is computing $u^{n\Delta t}$ using $n$ updates.
> >
> > > The authors remark that it is very easy to implement this method compared to the classical methods. Could the authors elaborate on this? How easy is this method? In what way? Usually, in a conventional code, the important pieces are the correct differentiation rules / stencils, communication between values in the volume and the values on the facets, and implementing the special adjustments at the boundaries, among many other aspects. How does NFV fare on these considerations?
> >
> > This method is easy to implement compared to complex method such as discontinuous Galerkin, which needs a careful implementation to work: details such as the way integrals are computed are extremely important to have a stable implementation. Our method is exactly as easy to implement as any finite volume method. However, some of those methods (eg. WENO) have more complex flux functions that needs a bit more attention than a neural network during implementation. One of the key points of the method is that it is straightforward to implement from a finite volume implementation since the only consideration is changing the flux function (and taking care of the training of course). The important parts of conventional code that you mention are still very important and are unchanged using our method.
> >
> > > Also, by the virtue of choice of the example PDEs, I am assuming that the authors did not need to calculate the spatial derivatives of the primal variables during the training process. That is, both the $t-$ and the $x-$ derivatives are integrated away, and the loss functional  does not contain any spatial derivatives of the function . Is that correct?
> >
> > This is correct. For supervised learning, this is straightforward since the equation is un-used. For unsupervised learning, the spacial derivative disappears using an integration by part. The time derivative is still computed but replaced by a discrete derivation using $\partial_t u(t, \cdot) \sim (u(t+\Delta t, \cdot) - u(t, \cdot)/\Delta t)$.

---

### Author Response · Authors · 2025-11-26
**Look forward to your feedback!**

Dear Reviewers,

Thank you again for your detailed feedback. We believe we have addressed all concerns raised in the reviews and discussion, but we haven't yet heard back from any reviewer. If there is anything further we can clarify or improve in the coming days that would help you feel confident in increasing your score, please let us know. We are happy to provide any additional material while there is still time before the deadline.

We appreciate your time and consideration.

---

### Author Response · Authors · 2025-12-03
**Authors Final Remarks for AC (1/2)**

Dear AC,

Below is a short summary of the consensus strengths and weaknesses, and how we addressed each concern.

### Main strengths noted by reviewers

- Well-written, clear problem formulation, ideas clearly conveyed.
- Novelty of the proposed method, solid empirical evaluation with convincing improvements over baselines.
- Proposed method achieves: a) up to 10x lower error than classical finite volume methods and matches DG (a complex modern finite-element method)'s accuracy but is much faster, practical and simple to implement, b) guarantee conservation of mass, incorportated into a neural network, c) supports both supervised and unsupervised learning, d) generalizes to more complex initial conditions despite being trained on simple Riemann problems building blocks.
- Solid ablation showing evolution of error vs discretization size.
- Extensive experiments across multiple PDEs as well as empirical study on experimental highway data is valuable, showing method generalizes and handles noisy conditions and is practical.

### Main concerns raised by reviewers

- Reviewers requested more detailed explanations in parts of the method (neural network design, hyperparamer tuning...) and discussion on the limitations and competitiveness of the method as well as additional experimental details (inference time, training time, memory consumption, etc).
- Reviewers raised concerns about missing comparisons with non-classical methods (e.g. PINNs in unsupervised case)
- Reviewers raised concerns about all synthetic experiments being on one-dimensional conservation laws.
- Reviewers requested new ablations on the CFL condition and on the architecture and dataset choices.

We have answered all reviewers' doubts and questions in detail in our responses, which have led to substantial improvements in the paper. **We have uploaded a revised PDF with all changes marked in blue.** **In our final remark below (2/2), we detail how we address all reviewer's concerns and requests, and which changes they led to in the paper.**

### Summary

We thank the reviewers for their careful evaluations. We believe the additions and revisions we have done to address the reviewer's concerns (as detailed in final remarks 2/2) led to a stronger paper. Some concerns included the absence of ML-method comparisons (we explained that many modern ML-based PDE solvers are not directly applicable to shock-dominated hyperbolic problems, making such comparisons nontrivial and outside the scope of this paper); the focus on one-dimensional problems (we clarified that 1D conservation laws are widely used and a standard and rigorous testbed for new numerical solvers, that we did an extensive and in-depth empirical study across seven PDE variants and experimental highway data and our focus on 1D allowed us to perform an unusually broad and in-depth empirical study across seven PDE variants, and that doing a proper study on 2D is a major piece of work); and the lack of theory (we explained that we submitted a journal paper in parallel containing 30+ pages of theoretical proofs about NFV's convergence, thus we focused this ICLR paper on a comprehensive empirical study showcasing NFV's accuracy, speed, robustness and stability, and we are willing to include high-level theorem statements in the appendix in the final version if it is deemed necessary).

The remaining concerns and how we addressed them is detailed below; in short, we conducted new trainings and included a new ablation, we included a lot more architectural and experimental details into the main body of the paper, we explained more carefully the points that reviewers struggled with or understood incorrectly, and we included more details about most concerns and questions raised by reviewers.

---

> ### Author Response · Authors · 2025-12-03
> **Authors Final Remarks for AC (2/2)**
>
> Following the reviews, we have uploaded a revised manuscript with all changes marked in blue. Below is a breakdown of the additions and modifications we made.
>
> (1) New ablation: Table 3 shows training and evaluation results for our method and every baseline as a function of the CFL condition for six values in [0.2, 1.2], showing our method still outperforms the baselines.
>
> (2) We added a lot of experimental details into the main text, including details about the model architecture, the synthetic and experimental datasets, as well as training time, memory requirements, and inference time, which we compare with existing schemes.
>
> (3) We added clarifications about many points reviewers had doubts on, raised questions about, or understood incorrectly:
> - We train on simple initial conditions viewed as elementary building blocks, and later evaluate on complex initial conditions comprising interacting weak and strong shocks and rarefactions. This is not a limitation: one could train directly on complex data. Rather, the fact that NFV can be trained solely on analytically tractable Riemann building blocks while still generalizing reliably to far richer piecewise-constant configurations is both practically advantageous and conceptually elegant.
> - We precised that (U)NFV still diffuses, but notably more slowly than standard finite-volume schemes. Sharp features dissipate less, leading to a more robust method.
> - (U)NFV is trained to solve one conservation law, with short training time, and can be reused for any number of initial conditions with various CFL conditions.
> - Our method is very competitive in terms of time to solve: once training optimization is done, inference is vectorized and very fast (O(T)); no optimization is done at inference time.
> - No spatial derivatives of primal variables are required during training with the weak loss: the integration by parts removes them. Time derivatives are handled via simple finite differences.
>
> (4) Regarding the reviewer's suggestion of comparing to additional non-classical methods: (i) our primary goal in this paper is to provide a fast, substantially more accurate drop-in replacement for classical finite-volume and DG solvers, which are used by practioners and in the industry, so we focus our experiments on those widely used baselines (including DG which is a SOTA method) rather than on methods that are not typically deployed on these problems, (ii) many popular neural approaches (PINNs, operator learners such as FNO/DeepONet, and recent hybrids) have been developed and mainly validated for smoother elliptic/parabolic PDEs, and are known to struggle or need significant modification in the shock-dominated conservation-law setting we consider, making a fair, apples-to-apples benchmark nontrivial, and (iii) conceptually, NFV largely differs from these more specialized methods by tightly integrating a small CNN with the finite-volume update so that conservation and much of the stability come from the underlying scheme, yielding simple, scalable architectures that are directly usable by practitioners already familiar with FV codes.
>
> (5) We have included explanations throughout the paper regarding the reviewer's concern about our work being on 1D PDEs: (i) starting in 1D is standard for new hyperbolic solvers and we believe scientifically sufficient here, since 1D conservation laws already lead to complex interacting shocks and are widely used in practice (traffic flow, channels, pipelines...), (ii) the core NFV design is not intrinsically one-dimensional and can be extended with FV updates on the faces of a 2D/3D volume, but a multi-D study (new equations, schemes, exact/benchmark solutions) is substantial work that we believe belongs to a subsequent paper, and (iii) in this paper we instead invest that space and compute budget into a large 1D evaluation, with seven distinct PDE variants, many numerical baselines, both supervised and unsupervised training cases, and large real-world traffic data, as well as modern vectorized implementations of all these, to give a thorough and reliable assessment of (U)NFV.
>
> (6) We added details on why our method is easier to implement and to run experiments with than a complex state-of-the-art method such as Discontinuous Galerkin, while achieving comparable accuracy and demanding significantly less compute time.
>
> (7) We included that NFV can adapt to various boundary conditions (eg. Dirichlet, Neumann, open boundaries...) using ghost cells of interface fluxes. Non-uniform and adaptive sampling strategies are also discussed.
>
> (8) We detailed the impact of the architecture choice and training data size, as well as the impact of the weak loss design (namely the chosen base function polynomials).
>
> Thank you for your time!
>
> Best regards,
>
> The authors

---

### Meta-Review · Area_Chair_ZzJP · 2026-01-03

**Summary:**

The paper received mixed scores from four reviewers. Below are the major concerns raised in their initial reviews:

1) The paper focused on 1D, moderately sized problems only. The proposed method’s extensions to higher-dimensional, larger-scale problems were unclear.

2) The comparisons were limited to classic methods only; comparisons with modern neural network baselines should be included.

3) The paper should provide more training details, e.g., training time, data requirements, and memory consumption, to help readers better evaluate the results.

Reviewers have also raised individual concerns about the network models' design decisions, the theoretical analysis of the method’s convergence properties, the method’s extensions to more challenging geometries/boundary conditions, and the paper’s view of its limitations.

**Reviewer Concerns:**

Regarding the first concern, the rebuttal claimed that work on the 1D case has already made significant contributions and acknowledged that extensions to 2D/3D would require nontrivial changes (which they plan to publish the 2D study in a separate paper). I think this would become a debatable topic in the discussion.

Regarding the second concern, the rebuttal claimed that comparisons with neural network baselines are difficult due to 1) many widely used network baselines are designed for smooth problems rather than the shock-dominant setting discussed in the paper; 2) the proposed method is conceptually different from these existing methods. I think this remains a debatable concern.

Regarding the last concern, the rebuttal clarified that the supplementary materials provide relevant details and summarized the key statistics. I think it has properly addressed this concern.

**Reviewer Scores:**

Most of the reviewers were mildly positive before the rebuttal. Many of them pointed out the lack of comparisons with network baselines and the limitation to 1D problems as weaknesses. As I summarized above, I am not sure whether these reviewers would be satisfied with the rebuttal’s arguments on these two weaknesses. Having more experiments would probably convince them more easily to support this paper.

The mildly negative reviewer was also concerned about the lack of comparisons with network baselines, but I think their other concerns could probably be addressed by the current rebuttal (basically, clarifications regarding the network model's design decisions and the statement of limitations). I think they would likely maintain a borderline score.

Overall, I think the reviewers would be more likely to converge on accepting this work rather than rejecting it. However, rejection is also possible if, after discussion, some reviewers did not buy 1) focusing on 1D problems is sufficient, and 2) lacking comparisons with the network baselines is acceptable.

---

### Decision · Program_Chairs · 2026-01-26

Accept (Poster)